# Simulation of Extreme Heatwaves with Empirical Importance Sampling

Pascal Yiou[1] and Aglaé Jézéquel[2]

[1]Laboratoire des Sciences du Climat et de l'Environnement, UMR 8212 CEA-CNRS-UVSQ, IPSL and U Paris Saclay, 91191 Gif-sur-Yvette cedex, France
[2]Laboratoire de Météorologie Dynamique, UMR CNRS-ENS-UPMC-X, IPSL and U Paris-Sorbonne, 75005 Paris, France

**Correspondence:** Pascal Yiou (pascal.yiou@lsce.ipsl.fr)

**Abstract.** Simulating ensembles of extreme events is a necessary task to evaluate their probability distribution and analyse their meteorological properties. Algorithms of importance sampling have provided a way to simulate trajectories of dynamical systems (like climate models) that yield extreme behavior, like heatwaves. Such algorithms also give access to the return periods of such events. We present an adaptation based on circulation analogues of importance sampling to provide a data-based algorithm that simulates extreme events like heatwaves in a realistic way. This algorithm is a modification of a stochastic weather generator, which gives more weight to trajectories with higher temperatures. This presentation outlines the methodology on European heatwaves and illustrates the spatial and temporal properties of simulations.

## 1 Introduction

The summer heatwaves in Western Europe in 2003 or in Russia in 2010 were not only record breaking events, but outliers of the temperature distribution by exceeding several standard deviations. Those events were considered as catastrophes in the countries where they occurred, and drastic measures of adaptation had to be taken, so that the following events (although milder) had much lower impacts. One can wonder whether such events were unprecedented because the observational records are too short (i.e. their return periods are longer than any time series of observations) or because climate change created new conditions of emergence. In order to solve this conundrum, it is necessary to be able to simulate the most intense event over a given region that is compatible with present-day conditions, and compare it with observed records. This necessitates efficient simulation methods. The general challenge that we want to address is to simulate an ensemble of heatwaves with a return period larger than 1000 years, with present-day conditions (i.e. less than 100 years of observations)

Available ensembles of climate model simulations from CMIP5 (Taylor et al., 2012), Euro-CORDEX (Vautard et al., 2013), and weather@home (Massey et al., 2015) contain mostly "normal" summers, so that samples of extreme summers are often scarce. In addition, model biases can add issues on the reliability of simulated structures or return levels (Maraun et al., 2017).

Long lasting events like heatwaves or cold spells yield a challenge beyond the simulation of large values of temperature. For example, heatwaves are characterized by prolonged episodes of high temperatures that are associated with persisting anticylonic atmospheric patterns (Cassou et al., 2005; Kornhuber et al., 2017; Quesada et al., 2012). The maximum duration of those events is obviously bounded by the seasonal cycle because the lower solar input in Autumn imposes an end to all summer heatwaves, although the length of seasons is subject to variations (Cassou and Cattiaux, 2016). This outlines the necessity to simulate the climate ingredients leading to a warm/cold spell. This inspired the idea of *storylines* that emphasize the mechanisms behind extremes (Hazeleger et al., 2015; Zappa and Shepherd, 2017; Shepherd et al., 2018; Shepherd, 2019) and investigate how they are affected by climate change.

Statistical models have been developed to simulate extreme events (Ghil et al., 2011). Extreme value theory (EVT) is useful to investigate and to simulate short-lived events, especially when they deviate from Gaussian distributions. But it might not be appropriate to investigate long lasting events (which end up leading to a Gaussian distribution). The investigation of multivariate fields (e.g. a temperature and the atmospheric circulation) is possible but requires rather complex implementations.

Dynamic weather generators that simulate ensembles of climate variables have been devised to circumvent this sampling difficulty. The weather@home experiment (Massey et al., 2015) simulate tens of thousands of trajectories of the HadAM3P model, which is the atmosphere component of the coupled ocean-atmosphere model of the UK Met Office Hadley Centre (Gordon et al., 2000). Although spectacular, this system is not very flexible (the model parameters are fixed, the region is imposed, etc.) nor optimal in the sense that one get "only" ten millennial heatwaves in $10^4$ runs. The limits of atmospheric only model approaches have also been demonstrated (Fischer et al., 2018; Dong et al., 2017).

Stochastic weather generators (Ailliot et al., 2015) are statistical models that yield reasonable physical features and can be run many times for a low computational cost. In principle, such stochastic models can enlarge the sampling size to millions of simulations so that one could obtain hundreds of extreme events with millennial return times rather inexpensively.

Ragone et al. (2017) were the first to perform simulations of extreme heatwaves in Europe with *importance sampling* algorithms and a simplified climate model. They were able to simulate a hundred of heatwaves with a return period of 1000 years at the cost of one hundred simulations, by avoiding simulating normal years. The main caveat of that proof of concept is linked to potential model biases (Fraedrich et al., 2005) and the lack of a seasonal cycle (they simulated perpetual summers).

The goal of this study is to assemble ideas from stochastic weather generators and importance sampling to simulate extreme events with realistic atmospheric circulation features. We will call this system an *empirical importance sampling algorithm*. For simplicity (and without lack of generality) we will focus on the simulation of summer European heatwaves, as a proof of concept. Hence the paper will be based on summer daily temperature in several European stations. The paper will present a data-based stochastic weather generator that nudges simulated trajectories toward high temperatures. We will investigate the physical and statistical properties of this weather generator. In particular, we will identify the weather types associated to extreme heatwaves.

Sec. 2 presents the temperature and atmospheric circulation data that are used in the paper. Sec. 3 recalls the ideas of importance sampling and details the analogue-based algorithm for sampling heatwaves. Sec. 4 shows the results of simulations of extreme heatwaves.

## 2 Data

### 2.1 Atmospheric circulation

We use the reanalysis data of the National Centers for Environmental Prediction (NCEP) (Kistler et al., 2001). We consider the geopotential height at 500mb (Z500) over the North Atlantic region for computation of circulation analogues. Sea-level pressure (SLP) is used for *a posteriori* diagnostics. We used the daily averages between January 1st 1948 and December 31st 2018. The horizontal resolution is $2.5°$ in longitude and latitude. The rationale of using this reanalysis is that it covers 70 years and is regularly updated.

We consider Z500/SLP fields over two regions outlined in Fig. 1. The region in red is used to compute Z500 analogues. It is similar to the one advocated by Jézéquel et al. (2018). The region in blue is used for verification. The reason to use Z500 for computations, rather than SLP, is linked to the "heat low" of SLP during heatwaves (Jézéquel et al., 2018). SLP and Z500 yield similar properties during the winter.

One of the caveats of this reanalysis dataset is the lack of homogeneity of assimilated data, in particular before the satellite era. This can lead to breaks in pressure related variables, although such breaks are mostly detected in the southern hemisphere and the Arctic regions (Sturaro, 2003), and marginally impacts the eastern north Atlantic region.

Since Z500 values depend on temperature, we detrend the Z500 daily field by removing a seasonal average linear trend from each grid point. This preprocessing is performed to ensure that the results do not depend on atmospheric trends. All the analogue computations of this paper were performed on detrended and raw Z500 data, in order to verify the robustness of the results to Z500 trends.

### 2.2 Temperature observations

We took daily averages (TG) of temperatures from the ECA&D project (Klein-Tank et al., 2002). We extracted data from Berlin, De Bilt, Toulouse, Orly and Madrid (Fig. 1). We consider data from June-July-August (JJA). Those five stations cover a large longitudinal and latitudinal range in western Europe. These datasets were also chosen because:

- they start before 1948 and end after 2018. This allows the computation of analogue temperatures with the Z500 from the NCEP reanalysis, which includes that period,

- they contain less than 10% of missing data.

These two criteria allow keeping 528 out of the 11422 ECA&D stations that are available in 2018.

The five daily temperature averages (TG) are then averaged in order to provide a daily European temperature index. This choice is done to simplify the presentation of results. Results for individual stations are presented in the supplementary material. This would overcome the potential caveat of the European temperature index, which does not capture the spatial variability of European heatwaves (Stefanon et al., 2012). Since we focus on extremely hot temperature spells, this proof-of-concept study is limited to summer, we consider the period between June 1st to August 31st.

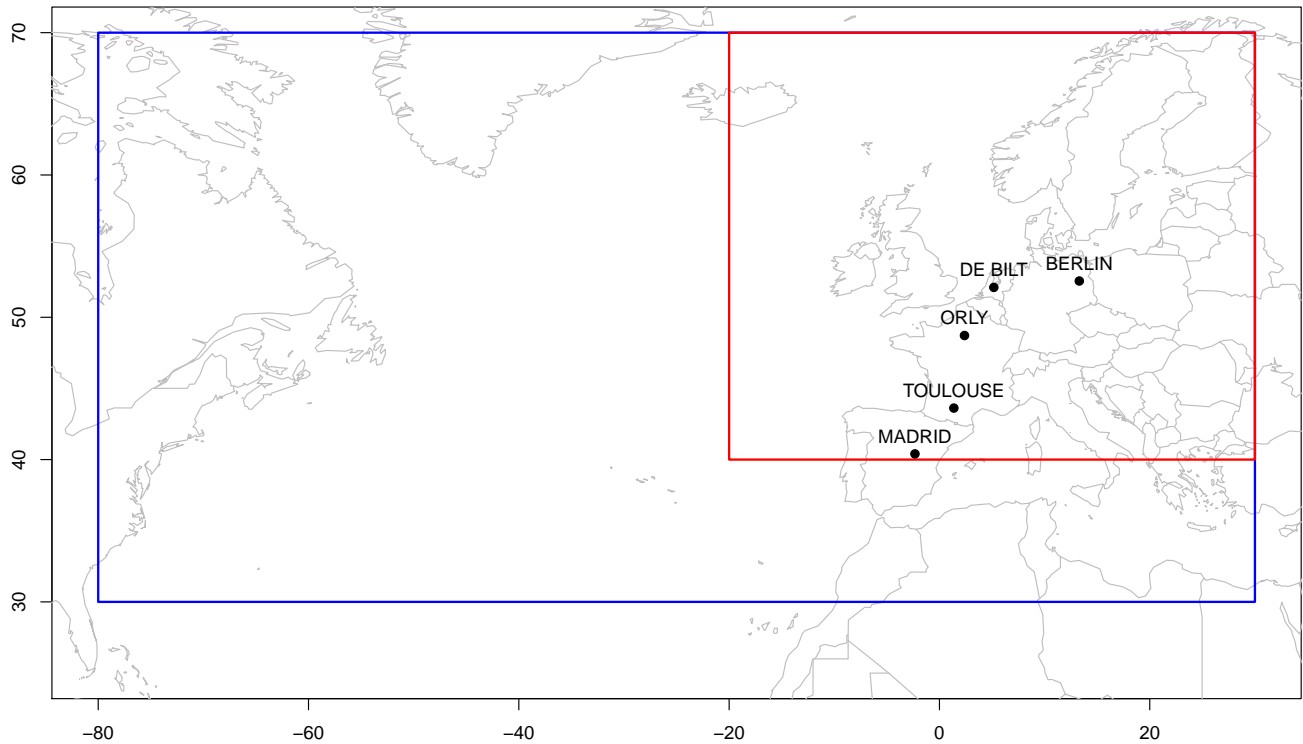

**Figure 1.** Location of five European stations. Longitudes are expressed in degrees east. Latitudes are expressed in degrees north. The red rectangle indicate the zone on which Z500 circulation analogues are computed. The blue rectangle indicates the region of pattern verification for Z500 and SLP.

## 3 Methodology

This section first recalls how analogues of circulation are computed and the general principle of a stochastic weather generator based on analogues. It then focuses on an empirical algorithm to simulate heatwaves, based on this stochastic weather generator.

### 3.1 Circulation analogues

5   Analogues of circulation are computed on Z500 data from reanalysis data of the NCEP reanalysis (Sec. 2.1) on the region outlined in red in Fig. 1. The reason to use this field rather than SLP is linked to the "heat low" of SLP during heatwaves (Jézéquel et al., 2018). SLP and Z500 yield similar properties during the winter. For each day between Jan. 1st 1948 and Dec. 31st 2018, the best 20 analogues (with respect to a Euclidean distance) in a different year and within 30 calendar days are searched. This follows the procedure of Yiou et al. (2013).

10   The small region (20°W–25°E; 40–70°N) is used to compute analogues of Z500, to simulate continental temperatures, following the domain recommendations of the analysis of Jézéquel et al. (2018). We compute analogues on "raw" daily Z500

data and detrended Z500. Detrending is performed by removing a smoothing spline on the spatially averaged Z500. The increasing trend of surface temperatures necessitates this trend removal in Z500.

The larger region (80°W–25°E; 30–70°N) covers the North Atlantic region. The atmospheric circulation of the North Atlantic evolves within this region. This region is used to compute composites of large-scale SLP/Z500 fields during simulated
events, for an evaluation of large scale features during those small scale events.

The analogues of Z500 are computed with the "blackswan" Web Processing Service (WPS) described by Hempelmann et al. (2018).

## 3.2   Stochastic weather generators

Ensembles of simulations of temperature can be performed with the rules illustrated by Yiou (2014), with analogue-based
stochastic weather generators (SWG). This type of SWG is equivalent to a resampling procedure (Ailliot et al., 2015).

A *static* SWG can be defined to simulate surrogate ensembles of Z500 sequences that are analogous to observed Z500 sequences. For each day $t$ between the 1st of June and the 31st of August, we keep the $K = 20$ best Z500 analogues. We randomly select one day $(k)$ among those $K$ analogues and $t$ (i.e., among $K + 1$ days), with a probability weight that is inversely proportional to the correlation of the $K$ analogues with the Z500 pattern at time $t$. This constraint favors analogues
with the best patterns, among those with the closest distance. This also favors the choice of $t$. With this type of SWG, simulated trajectories are random perturbations (by analogues) of an observed trajectory.

A *dynamic* SWG is defined to simulate ensembles of Z500 sequences that *could have been*, from a given initial condition. For an initial day $t$ (e.g., a 1st of June), we have $K$ best Z500 analogues. We randomly select one date $\tilde{t}$ among the dates of the $K$ analogues and $t$ (hence $K + 1$ dates), with a probability weight that is

1. inversely proportional to the number of calendar days between the analogues dates $\tilde{t}$ and $t$. This constrains the time of analogues to move forward. The weights of the analogues can be chosen proportional to $\exp\left(-\alpha_{cal}|\tilde{t} - t|\right)$, where $\alpha_{cal} \geq 0$ weighs the importance given to seasonality or the calendar day;

   2. proportional to the correlation of the analogue with the Z500 pattern at time $t$. This constraint favors analogues with the best patterns, among those with the closest distance.

The simulated next day $t'$ (e.g., a 2nd of June) is then the next day of the selected analogue: $t' = \tilde{t} + 1$ of the initial day (1st of June). Then $t$ is replaced by $t'$. This random selection of analogues is sequentially repeated until a lead time $T = 90$ days, to simulate a whole summer. This generates one random daily trajectory of Z500 or any climate variable between $t$ and $t+T$. Note that the dynamic weather generator spans a wider range of possibilities than the static weather generator, which is constrained by observed trajectories.

Those two types of SWG (static and dynamic) give useful information that are exploited in this paper. The random sampling procedures are repeated $S$ times to generate an ensemble of trajectories. Yiou (2014) showed that temperature biases are rather small but the time auto-covariance is under estimated in both SWGs.

### 3.3 Empirical importance sampling

The idea behind *importance sampling* is to simulate trajectories of a physical system that optimize a criterion in a computationally efficient way. Ragone et al. (2017) used such an algorithm to simulate extreme heatwaves with an intermediate complexity climate model. The procedure of importance sampling algorithms, say to simulate extreme heatwaves with a climate model, is to start from an ensemble of $S$ initial conditions and compute trajectories of the climate model from those initial conditions. An optimization *observable* is defined for the system. In this case, it can be the spatially-averaged temperature over Europe. The trajectories for which the observable (daily average temperature) is lowest during the first steps of simulation are deleted, and replaced by small perturbations of remaining ones. In this way, each time increment of the simulations keeps trajectories with the highest values of the observable. At the end of the season, one obtains $S$ simulations for which the observable (here average temperature over Europe) has been maximized. Since those trajectories are solutions of the equations of a climate model, they are necessarily physically consistent. Ragone et al. (2017) argue that the probability of the simulated trajectories is controlled by a parameter that weighs the importance to the highest observable values: if $n$ trajectories are deleted at each time step, the simulation of an ensemble of $T$-long trajectories has a probability of $(1 - n/S)^T$. Hence one obtains a set of $S$ trajectories with very low probability after $T$ time increments, at the cost of the computation of $S$ trajectories. For comparison purpose, if one wants to obtain $S$ trajectories that have a low probability ($p$) observable, then the number of necessary "unconstrained" simulations is of the order of $S/p$, so that most of those simulations are left out. Systems like weather@home (Massey et al., 2015) that generate tens of thousands of climate simulations are just sufficient to obtain $S = 100$ centennial heatwaves, and the number of "wasted" simulations is very high. Therefore, importance sampling algorithms are very efficient ways to circumvent this difficulty.

Here, we propose an adaptation of such an algorithm to the stochastic weather generators of Sec. 3.2 to simulate extreme heatwaves. The observable to be optimized is average daily temperature (named TG in the ECAD nomenclature (Klein-Tank et al., 2002)). To this end, we propose a new "rule" to the SWGs (static and dynamic) by giving more weight to the hottest temperatures. At each time step $t$, we note $t^{(k)}$ the dates of the $K$ best analogues. The temperature values of the $K$ analogues of $t$ and $TG_t$ are sorted in decreasing order. The ranks are written $R_k$ ($k \in \{0, \dots, K\}$). For example, the rank of the hottest temperature among analogues and temperature at day $t$ is 1. We chose weights $\left\{ w^{(k)} \right\}_{k \in \{0, \dots, K\}}$ so that:

$$w^{(k)} = A e^{-\alpha R_k}, \tag{1}$$

where $\alpha$ is a positive number and $A$ is a normalizing constant so that the sum of weights over $k$ is 1:

$$A = e^{-\alpha} \frac{1 - e^{-K\alpha}}{1 - e^{-\alpha}}. \tag{2}$$

The useful property of this formulation of weights is that the values of $w^{(k)}$ do not depend on time $t$, because the rank values $R_k$ are integers between 1 and $K + 1$. The weight values do not depend on the unit of the variable either, so that this procedure does not need major adaptation to simulate other types of climate variables (e.g. precipitation or wind speed). If $\alpha = 0$, this is equivalent to a stochastic weather generator described in Sec. 3.2.

The date for the next day is chosen at random by sampling $\{t, t^{(1)}, \ldots, t^{(K)}\}$ with the weights $\{w^{(k)}\}_{k \in \{0, \ldots, K\}}$. Therefore, if $\tilde{T}$ is the simulated temperature among $\mathbf{T} = \{\mathrm{TG}_t, \mathrm{TG}^{(1)}, \ldots, \mathrm{TG}^{(K)}\}$, then $\Pr\left(\tilde{T} = \mathbf{T}^{(k)}\right) = w^{(k)}$. The expected value $E(\tilde{T})$ is then:

$$E(\tilde{T}) = A \sum_{k=0}^{K} e^{-\alpha R_k} \mathbf{T}^{(k)} = A \sum_{k=0}^{K} e^{-\alpha k} \mathrm{sort}(\mathbf{T})^{(k)}, \tag{3}$$

where $\mathrm{sort}(\mathbf{T})$ are the sorted values of $\mathbf{T}$ in *descending* order. This allows to select the circulation analogues that favor the highest temperature. The $\alpha$ parameter gives some flexibility to select analogues with lower temperatures. It plays the same role as the simulation parameter of Ragone et al. (2017), which controls the return times of trajectories. This choice of weights is also interesting because it allows for an approximation of the mean value of simulations as a function of the parameter $\alpha$, as one can recognize a discrete Laplace transform of the distribution of $\mathbf{T}$ in Eq. (3).

This new rule replaces the analogue correlation weights of the dynamic and static SWGs defined in Section 3.2. The calendar day weight has to be maintained in order to keep a seasonality, especially for the dynamic scheme. Hence, this allows simulating

1. the hottest season that could have been with a similar atmospheric circulation, with a static analogue SWG,

2. the hottest season that could have been in the same climate, with a dynamic analogue SWG.

The algorithm is illustrated in Figure 2.

The dynamic SWG option is closest to the importance sampling experiments performed by Ragone et al. (2017). The major algorithmic difference is that importance sampling "eliminates" trajectories with "poor" properties of the observable, while the analogue importance sampling "favors" the trajectories with the "best" properties.

The analogue importance sampling uses a self coherent dataset as a basis (here: observations for temperature and a reanalysis for Z500 analogues) ensures that the simulated trajectories bear physical consistence between temperature and the atmospheric

circulation. The physical relevance also requires that simulations follow a seasonal cycle. If one is interested in simulating a perpetual summer, then the weight value $\alpha_{cal}$ on the calendar day is not very important. For physically realistic simulations, the value of $\alpha_{cal}$ can be chosen to be the smallest for which the temperature simulation median yields a seasonality. This prevents the system from staying in a perpetual mid-August.

The calendar "nudging" parameter $\alpha_{cal}$ needs to be chosen with care so that the additional rule does not create un-physical

simulations (i.e. with time going backwards or simulating a perpetual summer). The value of $\alpha_{cal}$ was estimated by trial and error, by taking the smallest value for which most (e.g. more than 70%) of the dynamic simulations end with dates after the second half of August. In our case (summer temperature simulations), a parameter value of 5 is deemed reasonable for summer temperature simulations. The value of $\alpha_{cal}$ could be different if another warm season has to be simulated, because the dynamics of temperature variations depends on the season.

From the numerical point of view, if $\alpha = 0.1$, 15 to 18 analogues have a probability larger than 0.1 of being selected. If seasonal trajectories ($N = 90$ days) are simulated, this means that more than $15^{90} \approx 10^{100}$ different trajectories of warm seasons are possible.

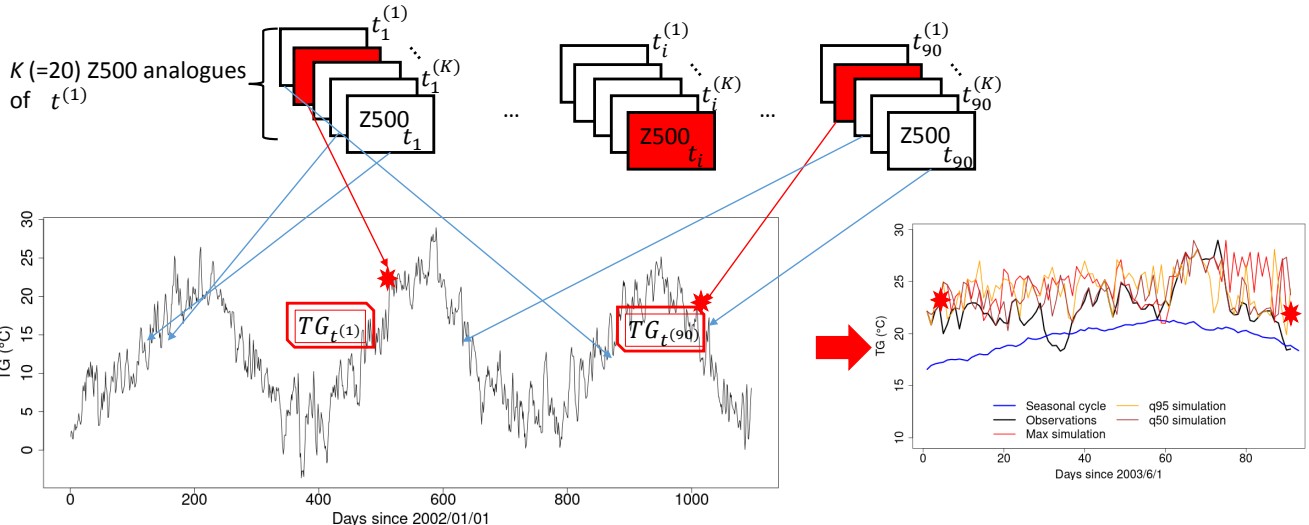

**Figure 2.** Illustration of the analogue importance sampling for mean daily temperature (TG). The red rectangles indicate the selected Z500 analogues for date $t^{(i)}$. The red stars in the lower graphs are the temperature values during the selected analogue date. Blue arrows indicate the correspondence between Z500 analogues and daily TG. Here, TG is the daily mean temperature for the Berlin-De Bilt-Orly-Toulouse-Madrid (BDOTM) average. The lower right panel illustrates simulated TG (red, orange and brown lines); the black line is the observed TG; the blue line is the seasonal cycle of TG.

From Eq. (3), the probability distribution of the simulations is linked to the value of $\alpha$. A formulation of the expected probability can be obtained heuristically. For example, let $Q$ be the smallest number so that:

$$\frac{1}{A}\sum_{k=1}^{Q}\exp\left(-\alpha k\right) > 1 - \epsilon, \tag{4}$$

where $\epsilon > 0$ is a small number (for example $1/N$, where $N$ is the number of simulations that are needed to observe one
5   simulated event) and $A$ was given by Eq. (2). Then the probability of dynamic trajectories with parameter value $\alpha$ is close
to $(Q/K)^M$, where $M$ is the average number of independent days during the simulated season ($M \approx 18$ for a season of 90
days), and $K$ is the number of analogues ($K = 20$). Such a heuristic formulation is close to what is obtained by Ragone et al.
(2017). Yet, this formulation suffers from numerical problems for large values of $\alpha$ (larger than 0.5) and small values of $K$
(like 20). An alternate empirical estimation of the probability distribution of the average of the trajectories is to consider that
10   the seasonal average of temperature closely follows a Gaussian distribution. Then an empirical estimate of the probability is

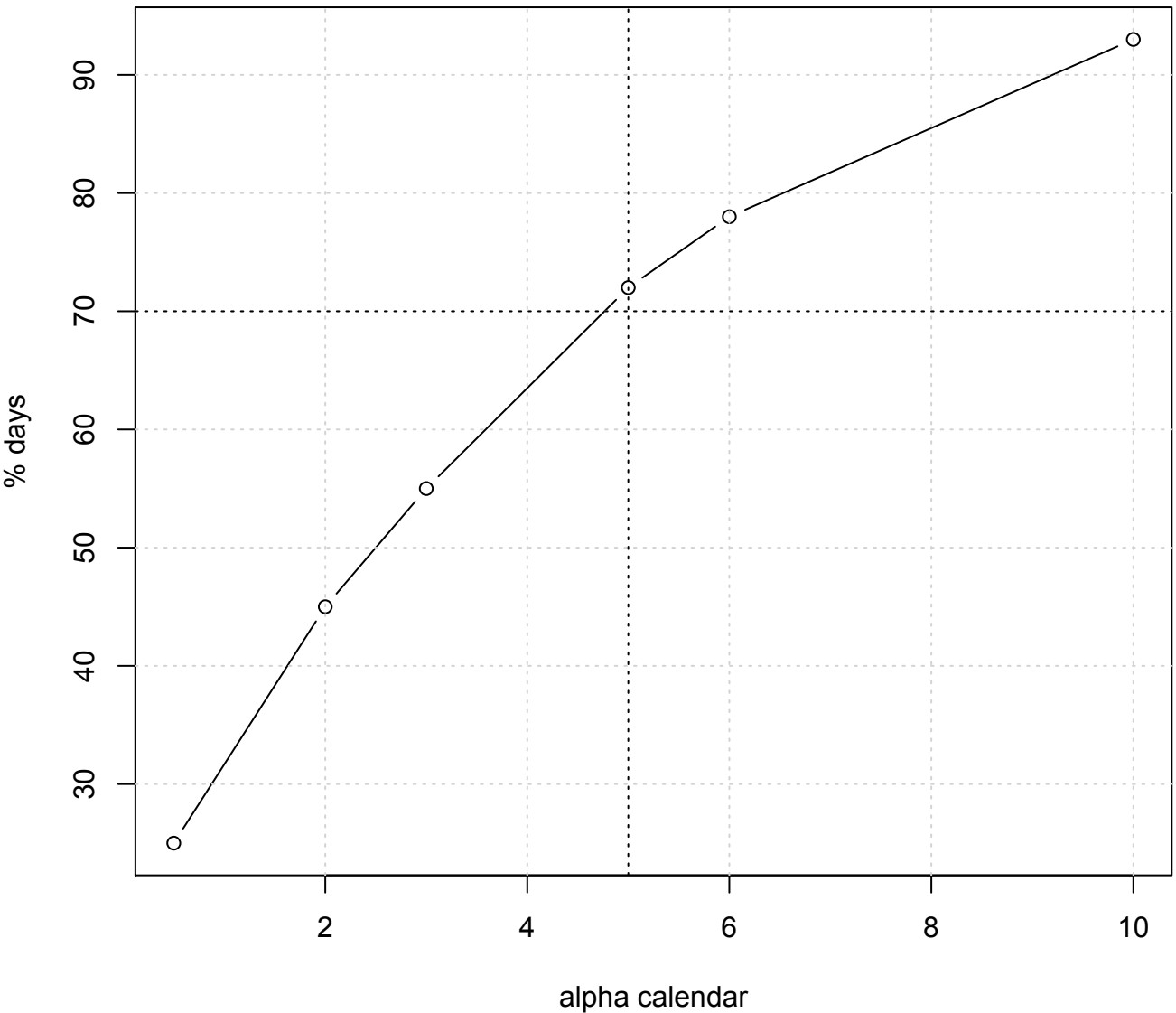

**Figure 3.** Percentage of dynamic simulations (from 100 simulations) of extreme summers for which the last day falls after August 15th, as a function of the parameter $\alpha_{cal}$, between 1 and 10. The vertical dashed line is for $\alpha_{cal} = 5$; the horizontal dashed line is for 70%.

obtained by comparing the quantiles of the observed distribution of average temperatures and the mean of simulated averages. With a chosen value of $K = 20$, the heuristic and empirical approximations give similar estimates of the probability (or return period) for values of $\alpha \leq 0.5$. We shall keep the empirical estimate of probabilities/return periods in the rest of the paper. Refined heuristic estimates are left for future (more statistical) work.

We emphasize that the methodology shown in this section (or in Ragone et al. (2017)) is quite general and not specific to summer heatwaves, as other types of extremes like cold spells (Cattiaux et al., 2010) or long episodes of precipitation (Schaller et al., 2016) can be envisioned.

## 4    Results

SWG simulations are done for each individual city (Berlin, De Bilt, Orly, Toulouse and Madrid) and their average. Only the
results on the average over the five cities are shown in the core of the paper, for compactness. Results for individual cities are shown in the Supplementary Figures. $S = 100$ simulations are performed with the static and dynamic SWGs. The simulations are initialized by conditions at the 1st of June, between 1948 and 2018, and are run until the 31st of August. Hence $100 \times 71$ summers are simulated for each experiment. The parameter $\alpha$ takes its values in $\{0, 0.2, 0.5, 1\}$ to evaluate the relation between the simulation averages and their return period (or probability).

In order to evaluate the effect of the initial condition on the simulated patterns, we select the warmest (2003), coldest (1956) and median (1986) summers for the averaged European temperature. For illustration purposes, we add 2018, which was a major heatwave in Northern Europe. This selection helps showing how extremely hot summers can be amplified (2003 and 2018), and how median and rather cool summers could have been hot, with present day climate conditions.

### 4.1    Interannual trends

The distribution of temperature simulations are shown in Figure 4, with parameter $\alpha$ values of 0, 0.2, 0.5 and 1. The static SWG distributions (blue boxplots) yield variations that are closely correlated ($r = 0.75$) with the observed values (black lines). The dynamic SWG distributions (red boxplots) have lower amplitude variations, although they are also correlated with observed values ($r = 0.58$). Figure 4 shows that the SWG distribution means increase with the $\alpha$ parameter. When $\alpha = 0$, there is no importance sampling, so that both types of simulations fluctuate around the mean value of observations (Figure 4a). In that
case, the record value (in 2003) or second highest value (in 2018) of temperature is not reached by the $S = 100$ simulations. The record low temperature (in 1956) is also barely reached.

We observe no trend in the simulated trajectories with $\alpha > 0$ for all years. This is explained by a tendency to select recent analogues in the importance sampling, because of the general temperature trend. The amplitude of the simulations is higher for dynamical simulations than for static simulations, because the simulated trajectories are not constrained by the observed ones,
and could hence behave in a rather different way.

As soon as $\alpha > 0$, the distributions of simulated temperature exceed the observed values (Figure 4b-d), because the SWG tend to select analogues for which temperature is higher than observed, by construction. When $\alpha > 0$, the dynamic trajectories

generally yield higher values than the static trajectories, because the dynamic SWG chooses the warmest analogues among all analogues. We observe that when $\alpha \geq 0.5$ then the dynamic trajectories are all above the 2003 record. Roughly half of the static trajectories with $\alpha = 1$ stand above the 2003 record.

When $\alpha > 0$, the static SWG generates similar circulation patterns that lead to warmer summer temperatures (blue boxplots). The 2003 record value in observations always yields the warmest JJA simulations. 2018 is the second hottest summer in the observations but it is no longer the second when $\alpha > 0$, with the static SWG, as a few years with heatwaves with analogue atmospheric circulation could have exceeded the 2018 temperature value.

We observe an increase of average extreme summers with parameter $\alpha$. Fig. 5 summarizes the temperature probability distributions for all years, in the observations and simulations, for varying values of $\alpha$. Here, return periods are estimated from a Gaussian approximation of the variability of average temperature (e.g. the black boxplot in Fig. 5) for present-day conditions. This is justified by noting that we consider the probability distribution of temporal averages (92 days in JJA) of spatial averages (5 stations). Those averages motivate a Gaussian approximation from the Central Limit Theorem (von Storch and Zwiers, 2001).

Here we obtain a return time of the 2003 summer heatwave of around 350 years in present-day conditions (i.e. the last decades) (Fig. 5). The return time of this heatwave has been estimated to be as large as $10^6$ years (Schaer et al., 2004; Stott et al., 2004; Cattiaux and Ribes, 2018). Such results were based on extrapolating the probability density law of temperature observations since 1900. Therefore the underlying hypothesis to estimate the return period (or probability) is different from the estimate of Schaer et al. (2004). The SWG approach answers the question "how likely is the occurrence of an event (in present day conditions)?" Note that the event itself does not need to have occurred. Using long time series answers the question "what is the frequency of the event?" In that case, the event must have occurred. Hence, our estimate is closer to a "hitting time" than a "return time", according to the definition of Haydn et al. (2005).

For the rest of the paper, we chose a value of $\alpha = 0.5$, which corresponds to heatwaves whose intensity is comparable to the 2003 record for the static SWG, and higher for the dynamic SWG (Fig. 5). The simulated heatwaves are more intense than the one of 2003, although the climate conditions are similar to present day ones. Therefore, such heatwaves are even rarer than the 2003 record value.

For comparison purposes, we simulated local heatwaves in each of the five stations. The appendix figures A1, A3 and A5 show the probability distribution of simulated summers (with $\alpha = 0.5$) for Berlin, Orly and Madrid.

## 4.2 Daily variations

The daily variations of dynamic simulations of average temperature for the four selected years (1956, 1986, 2003 and 2018) are shown in Fig. 6, with $\alpha = 0.5$. This figure illustrates how the simulated temperatures evolve above the seasonal cycle.

The values exceed the observed values within 3 days, especially for the coldest summer (in 1956, Fig. 6b). The maximum of the ensembles (red lines) generally reaches a plateau in 10 days after June 1st. The median and 95th quantile trajectories yield a higher temporal variability, which is lower than the observed one. We note that the algorithm allows selecting colder Z500

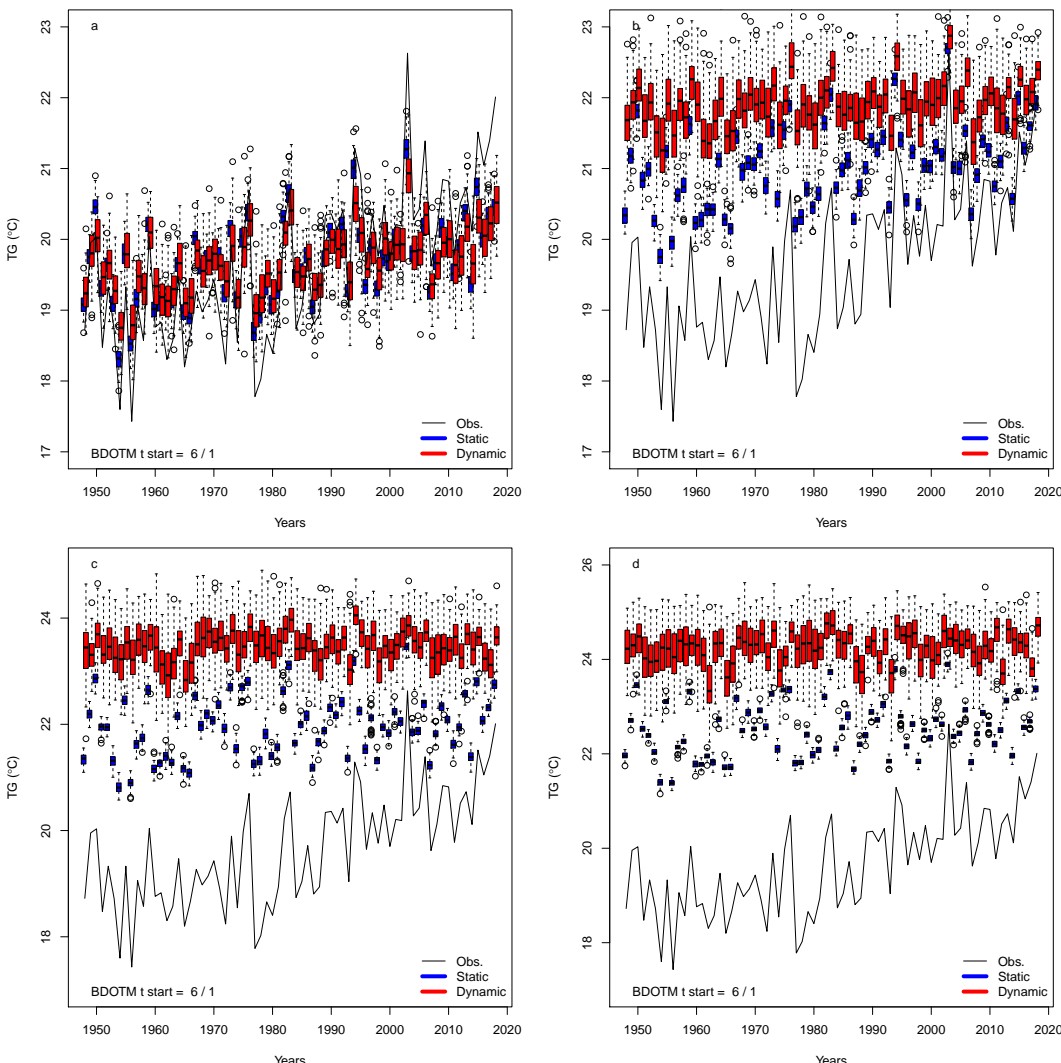

**Figure 4.** Time variations of probability distributions of simulated average temperatures (TG in °C) for four values of the weight on temperatures $\alpha_{TG}$: $\alpha_{TG} = 0$ (a). $\alpha_{TG} = 0.2$ (b). $\alpha_{TG} = 0.5$ (c). $\alpha_{TG} = 1$ (d). The black continuous line represents the observed variations of Berlin-De Bilt-Orly-Toulouse-Madrid (BDOTM) summer averages between 1948 and 2018. The vertical colored lines outline the coldest (blue), median (green), warmest (red) and 2018 summers. The boxplots represent the ensemble variability of the simulations for each year. The red boxplots are for the dynamic simulations. The blue boxplots are for the static simulations. The boxes of boxplots indicate the median ($q50$), lower ($q25$) and upper ($q75$) quartiles. The upper whiskers indicate $\min[\max(T), 1.5 \times (q75 - q25)]$. The lower whisker has a symmetric formulation. The points are the simulated values that are above or below the defined whiskers.

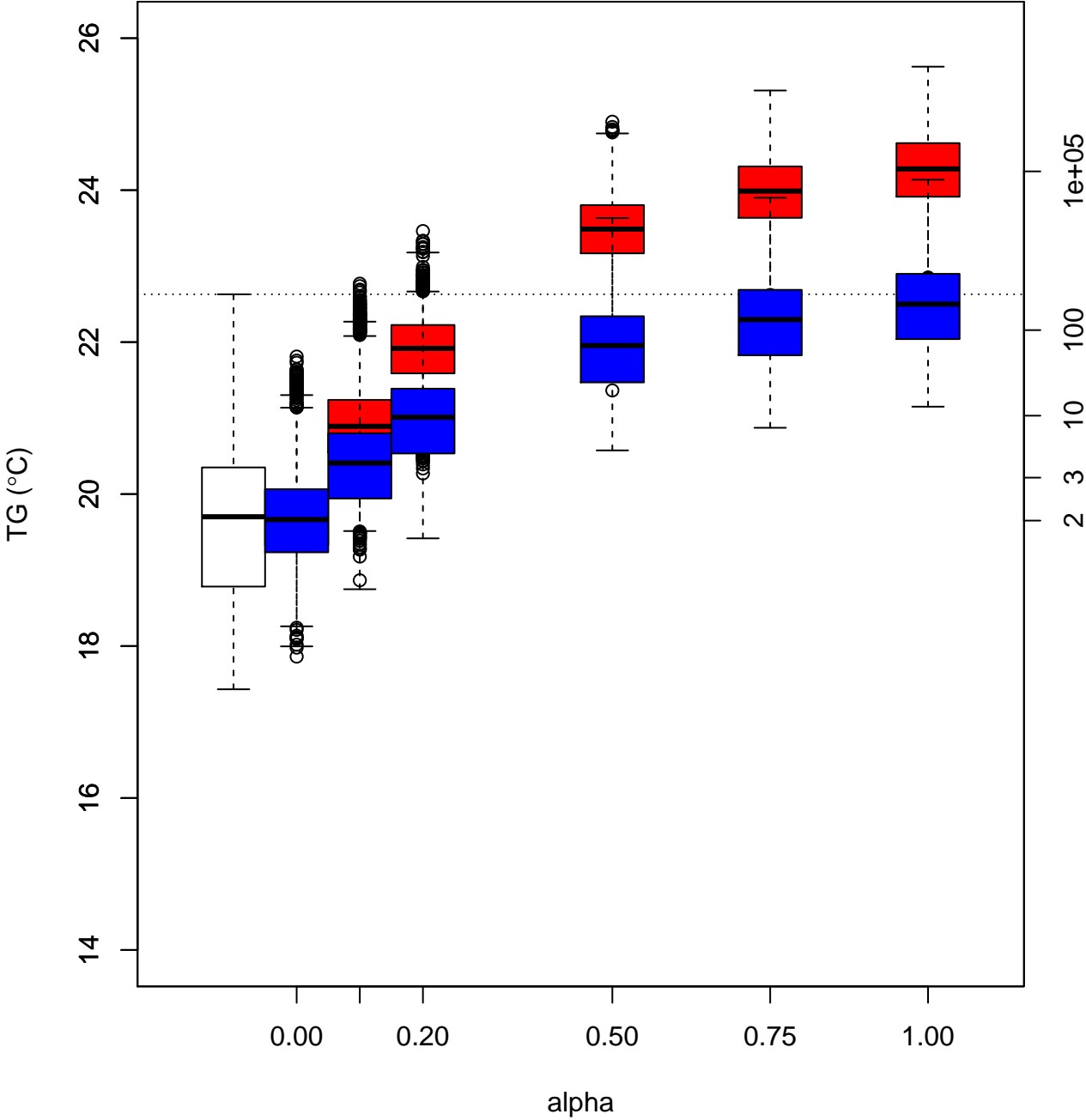

**Figure 5.** Boxplots of simulated June-July-August (JJA) temperatures (TG in °C) as a function of the parameter $\alpha$ for all years. The boxes of boxplots indicate the median ($q50$), lower ($q25$) and upper ($q75$) quartiles of the simulated values. The upper whiskers indicate $\min[\max(T), 1.5 \times (q75 - q25)]$. The lower whisker has a symmetric formulation. The points are the simulated values that are above or below the defined whiskers. The axis on the right indicates return times (in years), assuming a Gaussian distribution of JJA temperature averages with parameters estimated from the white boxplot. The horizontal dotted line is the mean TG in 2003.

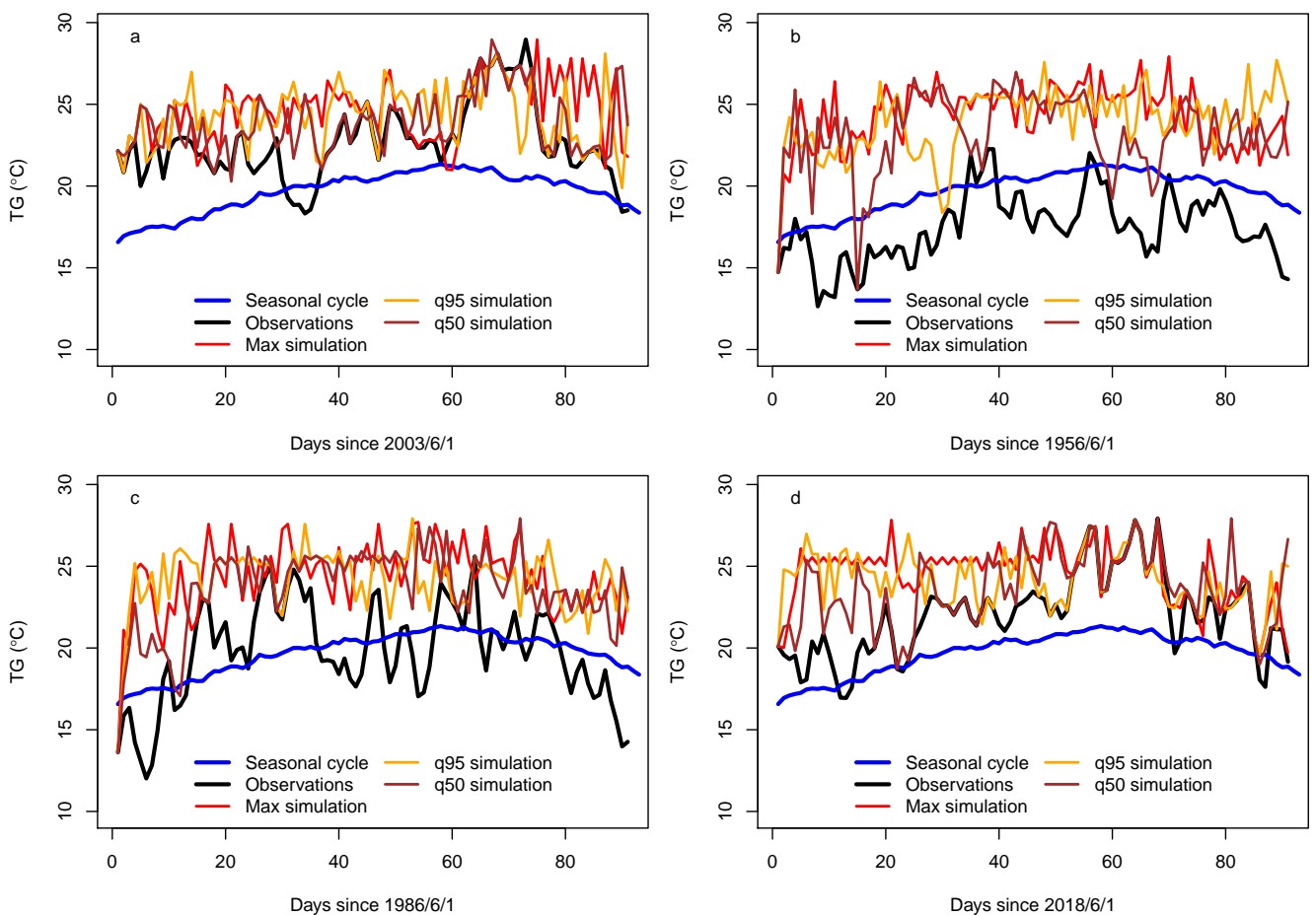

**Figure 6.** Observed and simulated daily temperature (TG) variations (for the Berlin-De Bilt-Orly-Toulouse-Madrid (BDOTM) average) for the four selected summers: 2003 (a), 1956 (b), 1986 (c) and 2018 (d). Temperatures are expressed in °C. The simulation parameters are $\alpha_{cal} = 5$ and $\alpha = 0.5$. The blue lines indicate the average seasonal cycle. The black lines are the observed TG. The red lines are the simulations with the highest JJA mean TG. The dashed lines are the simulations with the 95th quantile of JJA mean TG. The brown lines are for the median simulations.

analogues, so that temperature does not continuously increase, but yields time variability. This also means that all trajectories are different from each other.

The daily variations also explain the weak (visual) correlation between the means of simulations and observations in Fig. 4. The whole correlation is constrained by the few days after the initial condition (1st of June), as the simulations progressively "forget" the initial condition, especially for the cold summer in 1956.

## 4.3 Atmospheric patterns

We focus on the atmospheric circulation patterns that prevail in summer (JJA) over the North Atlantic region for simulation ensembles, during the four selected years. We show the SLP (Fig. 7) and Z500 (Fig. 8) composites over JJA. The composites are the JJA averages for reanalyses and, over all static and dynamic simulations.

The observed mean JJA SLP patterns are very different for each year (Fig. 7a–d). In particular, the two major heatwaves of 2003 (hottest) and 2018 (second hottest) shown in Fig. 4 yield rather contrasting SLP and Z500 pattern anomalies at the North Atlantic scale, although both years are characterized by positive SLP/Z500 anomalies over Western Europe.

Static simulations show mean JJA SLP and Z500 patterns that are similar to the observed ones for 2003, 1956, 1986 and 2018 (Fig. 7e–h and Fig. 8e–h). The mean SLP/Z500 patterns show deepened or shifted structures to maximize mean temperature. Those figures illustrate how a slight modification of the atmospheric structure could increase European summer temperature. Therefore, small perturbations of the daily atmospheric Z500 structures (a few meters) can be associated to $\approx 4\mathrm{K}$ to the mean JJA temperature of the coldest summer in the time series (1956), with $\alpha = 0.5$. This temperature change is larger than the expected increase from a perfect gas law ($\approx 0.4$ K). Interestingly, the shifts in SLP patterns are much larger than for Z500.

Dynamic simulations simulate optimal atmospheric patterns that are very similar across the North Atlantic, with strong anticyclonic patterns over western Europe (Fig. 7i–l and Fig. 8i–l). The resulting SLP and Z500 patterns are very similar for the four selected years, with positive anomalies above western Europe. The Z500 conditions for 2018 contrast with the other years with positive anomalies in the central North Atlantic. This shows how the dynamic simulations differ from each other and depend on the initial conditions. The sensitivity to initial conditions was exploited by Yiou and Déandréis (2019) for ensemble forecasts with analogues.

The differences between the static and dynamic simulations illustrate the different concepts that those two weather generators convey. The dynamic simulations end up with rather similar ranges of temperatures, and resembling SLP/Z500 structures over the whole North Atlantic (high pressure or Z500 over western Europe), which can be interpreted as an optimal pattern leading to major heatwaves.

For comparison purposes, Fig. A2, A4 and A6 show the atmospheric Z500 patterns linked to local heatwaves centered on Berlin, Orly and Madrid (from East to West). The optimal Z500 patterns with dynamic simulations for Orly are similar to the ones of the average temperature across the five stations.

## 5 Conclusions

This paper presents a method to simulate ensembles of extreme climate conditions. The underlying principle was to combine ideas from importance sampling (Ragone et al., 2017) and stochastic weather generators based on circulation analogues (Yiou, 2014). This method was tested to simulate European summer heatwaves, but can be adapted to simulate other types of events. The stochastic analogue sampling ensures a physical coherence of daily variations (circulation and temperature) through realistic circulation patterns (Yiou, 2014). The main caveat of this method is that it is based on a closed framework of a relation between temperature and the atmospheric circulation. It does not take into account the role of feedbacks, like soil moisture

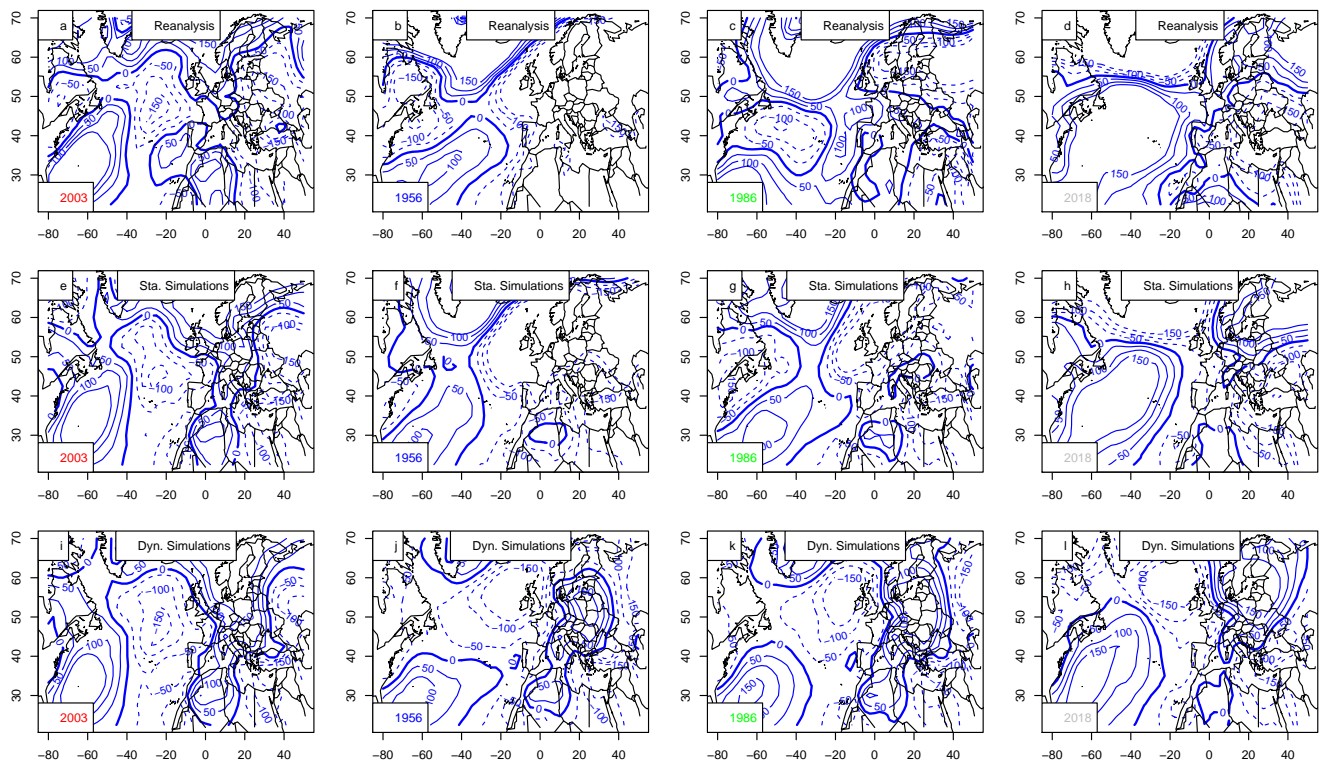

**Figure 7.** Maps of SLP anomaly composites (in Pa, with 50Pa increments) for the warmest summer (2003), coolest (1956), median (1986) and 2018. The 0 hPa isoline is indicated with a thick blue line. Negative anomalies are in dashed isolines. Horizontal axes are for longitudes in degrees east. Vertical axes are for latitudes in degrees north. Upper row (a–d): mean SLP from NCEP reanalyses. Center row (e–h): Static simulations. Bottom row (i–l): Dynamic simulations. The blue isolines are shown with 50 Pa increments. The thick blue line is for the 0 isoline. Dotted lines are for negative anomalies. Continuous lines are for positive anomalies.

(Vautard et al., 2007), that could amplify a temperature response. A second caveat is that this resampling methodology is useful to simulate seasons (or long periods of time). It would not be relevant to simulate short lived events, because there would not be enough resampling possibilities. We emphasize that daily temperature values are bounded by the observations, therefore one cannot simulate a daily value that has not been observed. However, the seasonal averages are not bounded because they are close to a Gaussian distribution, and simulated average variables can exceed (by far) the observed ones. The methodology explicitly exploits this mathematical property of random variables, in particular in the estimation of return periods. A third caveat of the present study is the length of available observations for the simulation of extreme events, so that the links between the atmospheric circulation and temperature might not be completely sampled.

This methodology simulates ensembles of extreme heatwaves that are possible in present-day conditions. We emphasize that no hypothesis on climate change is made to simulate events that are more intense than the record of 2003.

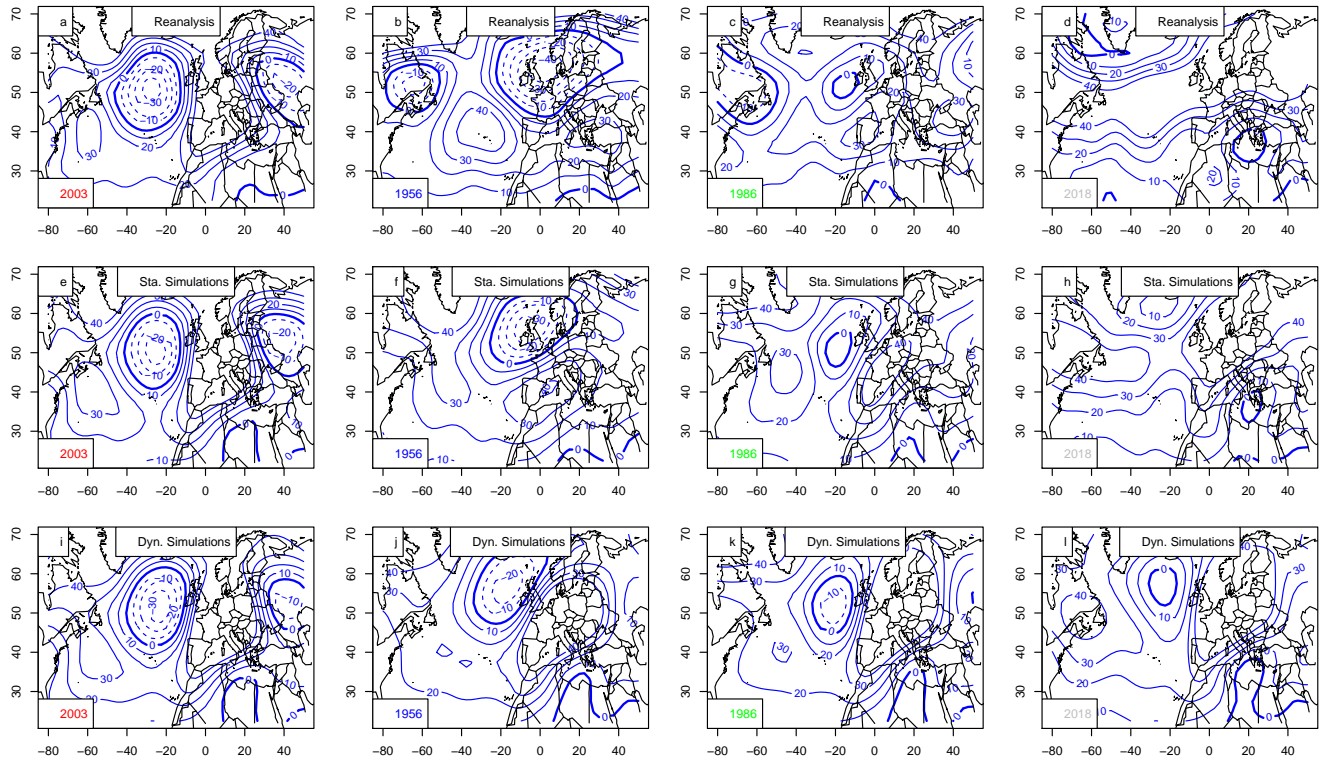

**Figure 8.** Z500 anomaly composites (in m, with 10m increments) for the warmest summer (2003), coolest (1956), median (1986) and 2018, for European averages. The 0 m isoline is indicated with a thick blue line. Negative anomalies are in dashed isolines. Horizontal axes are for longitudes in degrees east. Upper row (panels a–d): mean Z500 from NCEP reanalyses. Center row (panels e–h): Static simulations. Bottom row (panels i–l): Dynamic simulations. The blue isolines are shown with 10 m increments. The thick blue line is for the 0 isoline. Dotted lines are for negative anomalies. Continuous lines are for positive anomalies.

One parameter of the simulations controls the weight to be given to the hottest analogues. This parameter also controls the return period of simulation ensembles, similarly to the control parameter of Ragone et al. (2017). Thus, the simulated ensembles are associated to a return period (or a range or return periods).

This methodology is relatively easy to implement, and does not require to run a high resolution climate model, because it uses already existing datasets. It was tested on rather restricted cases of summer warm temperatures. It can be adapted to simulate other types of extreme events:

– Temperature extremes in other seasons. For example, extremely hot/cool winters or cool summers also have large impacts on society (energy, agriculture) and ecosystems' phenology.

– Extremes with other climate variables (precipitation, wind speed). Long lasting precipitation episodes do depend on atmospheric circulation patterns (e.g. Schaller et al., 2016). The fact that the weights that are chosen in the simulations do not depend on the units of the variables facilitates such an adaptation.

– Time-varying constraints can be added to simulate impacting compound events (Zscheischler et al., 2018), e.g. wet springs and hot summers. For example, one could maximize precipitation rate during spring, then temperature in early summer in order to generate events that have high impacts on the agriculture.

– Extremes in scenario model simulations. The examples of this paper exclusively used NCEP reanalysis (Kistler et al., 2001) data and ECAD (Klein-Tank et al., 2002) observations, for present-day climate conditions. Simulations could use the CMIP model database (Taylor et al., 2012) for analogue computation, and hence it would be possible to investigate changes in extremes for scenario simulations, relative to control simulations.

Therefore, we consider that this paper paves the way for many types of studies of impacts of extreme events and risk assessment for extremes (Sutton, 2019).

*Code availability.* The R code for simulations are available from Yiou (2019) under a free CeCILL licence. The analogues are computed with the "blackswan" WPS (Hempelmann et al., 2018).

*Data availability.* NCEP reanalysis and ECA& D data are available on open web sites. Sample input temperature and Z500 analogue files are provided by Yiou (2019).

*Sample availability.* A sample file for Orly temperature simulations is available from Yiou (2019).

## Appendix A: Diagnostics for Berlin

Time variations of simulated trajectories for extreme heatwaves (with $\alpha_{TG} = 0.5$) in Berlin are shown in Fig. A1. The hottest year is 1992. The coolest summer is in 1954. The median summer is in 1989.

The atmospheric Z500 patterns for the summers of 1954, 1989, 1992 and 2018 are shown in Fig. A2. The patterns are different from the ones that are obtained when maximizing European average temperature (Fig. 8).

## Appendix B: Diagnostics for Orly

Time variations of simulated trajectories for extreme heatwaves (with $\alpha_{TG} = 0.5$) in Orly are shown in Fig. A3. The hottest year is 2003. The coolest summer is in 1956. The median summer is in 1964.

The atmospheric Z500 patterns for the summers of 1956, 1964, 2003 and 2018 are shown in Fig. A4. The patterns are similar to the ones that are obtained when maximizing European average temperature (Fig. 8).

**Appendix C: Diagnostics for Madrid**

Time variations of simulated trajectories for extreme heatwaves (with $\alpha_{TG} = 0.5$) in Madrid are shown in Fig. A5. The hottest year is 1992. The coolest summer is in 1954. The median summer is in 1989.

The atmospheric Z500 patterns for the summers of 1977, 1982, 2015 and 2018 are shown in Fig. A6. The patterns are different from the ones that are obtained when maximizing European average temperature (Fig. 8).

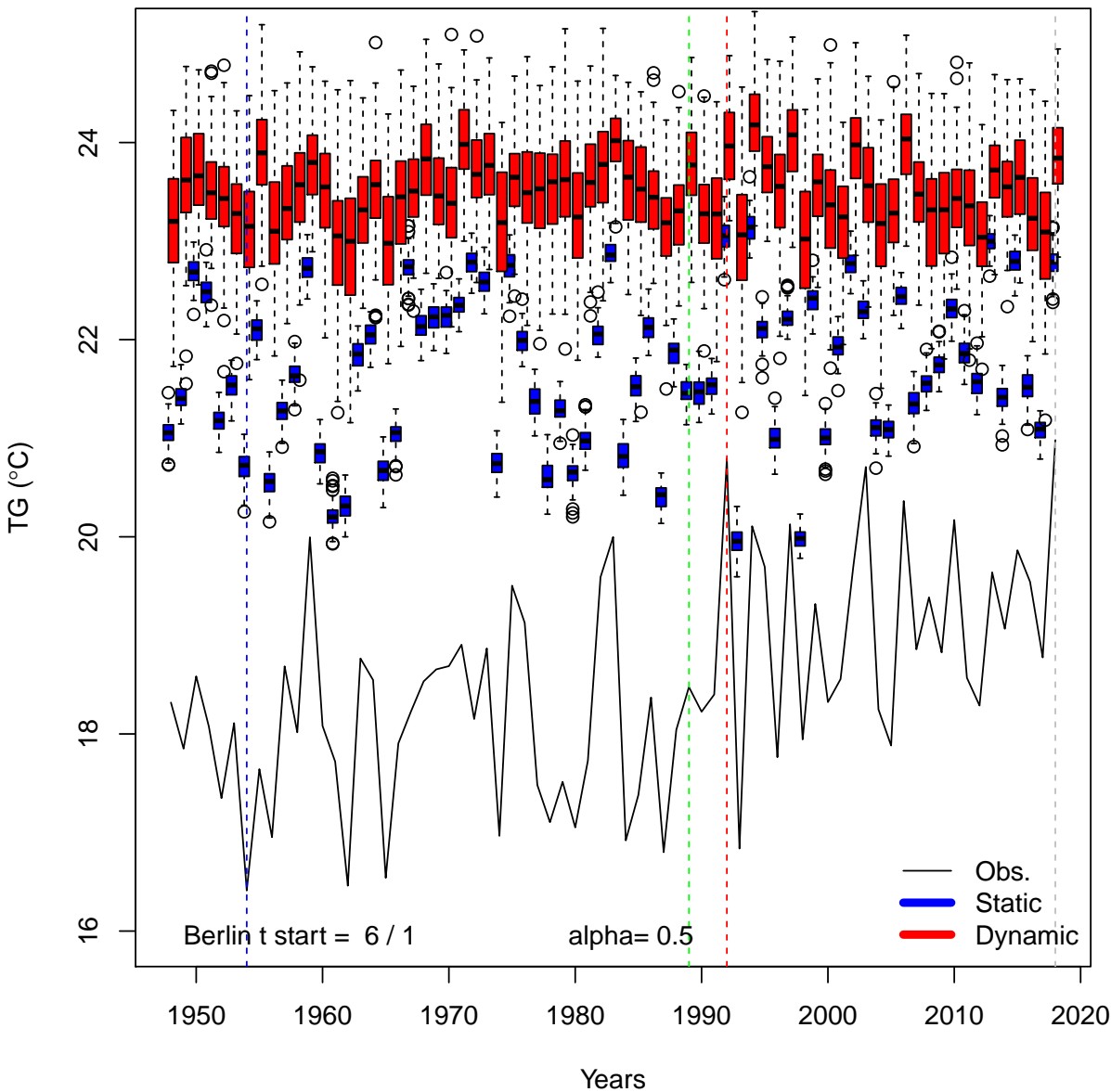

**Figure A1.** Time variations of probability distributions of simulated average temperatures (TG) for $\alpha_{TG} = 0.5$ in Berlin. The black continuous line represents the observed variations of Berlin summer averages between 1948 and 2018. Temperatures are expressed in °C. The vertical colored lines outline the coldest (blue), median (green), warmest (red) and 2018 summers. The boxplots represent the ensemble variability of the simulations for each year. The red boxplots are for the dynamic simulations. The blue boxplots are for the static simulations. The boxes of boxplots indicate the median ($q50$), lower ($q25$) and upper ($q75$) quartiles. The upper whiskers indicate $\min[\max(T), 1.5 \times (q75 - q25)]$. The lower whisker has a symmetric formulation. The points are the simulated values that are above or below the defined whiskers.

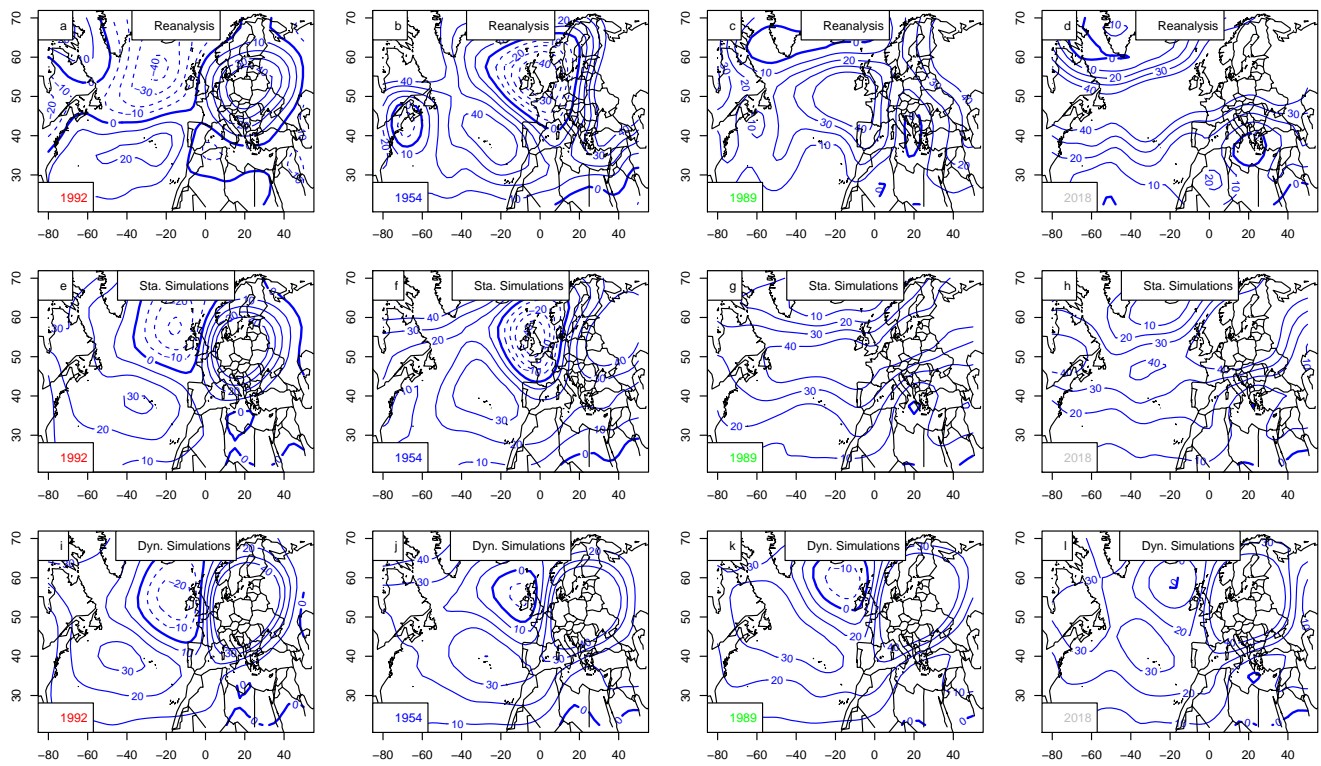

**Figure A2.** Z500 anomaly composites (in m, with 10m increments) for the warmest summer in Berlin (1992), coolest (1954), median (1989) and 2018. The 0 m isoline is indicated with a thick blue line. Negative anomalies are in dashed isolines. Horizontal axes are for longitudes in degrees east. Vertical axes are for latitudes in degrees north. Upper row (a–d): mean Z500 from NCEP reanalyses. Center row (e–h): Static simulations. Bottom row (i–l): Dynamic simulations.

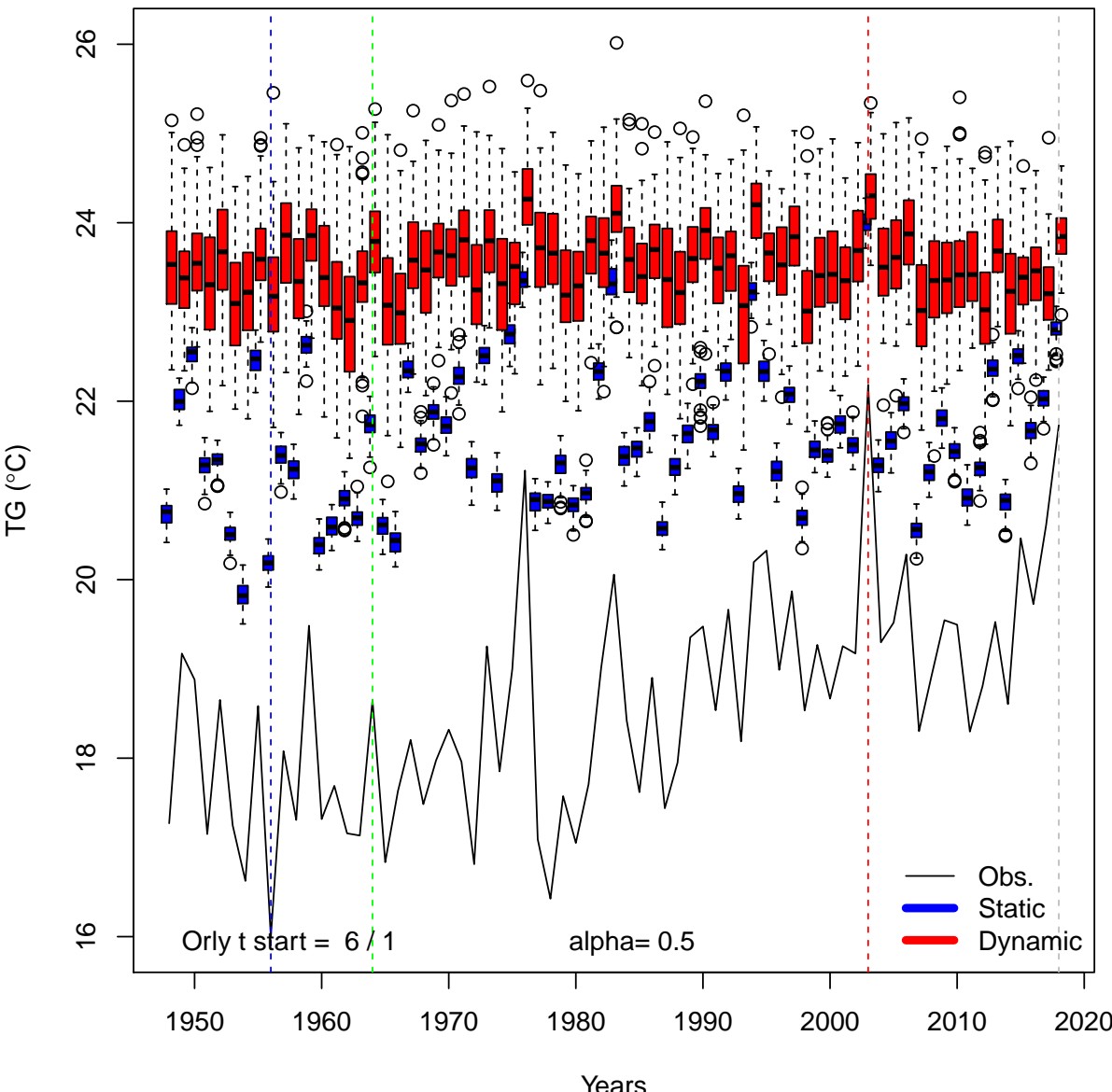

**Figure A3.** Time variations of probability distributions of simulated average temperatures (TG) for $\alpha_{TG} = 0.5$ in Orly. The black continuous line represents the observed variations of Orly summer averages between 1948 and 2018. Temperatures are expressed in °C. The vertical colored lines outline the coldest (blue), median (green), warmest (red) and 2018 summers. The boxplots represent the ensemble variability of the simulations for each year. The red boxplots are for the dynamic simulations. The blue boxplots are for the static simulations. The boxes of boxplots indicate the median ($q50$), lower ($q25$) and upper ($q75$) quartiles. The upper whiskers indicate $\min[\max(T), 1.5 \times (q75 - q25)]$. The lower whisker has a symmetric formulation. The points are the simulated values that are above or below the defined whiskers.

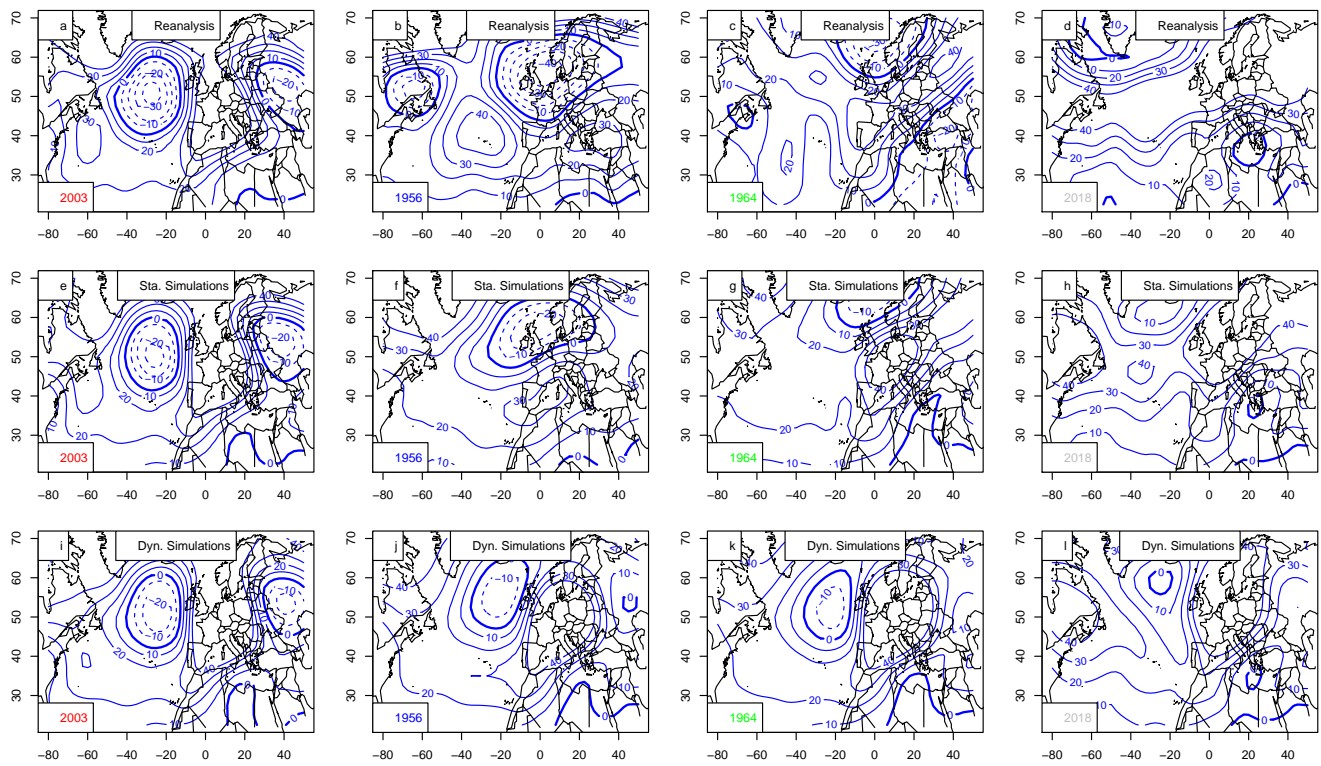

**Figure A4.** Z500 anomaly composites (in m, with 10m increments) for the warmest summer in Orly (2003), coolest (1956), median (1964) and 2018. The 0 m isoline is indicated with a thick blue line. Negative anomalies are in dashed isolines. Horizontal axes are for longitudes in degrees east. Vertical axes are for latitudes in degrees north. Upper row (a–d): mean Z500 from NCEP reanalyses. Center row (e–h): Static simulations. Bottom row (i–l): Dynamic simulations.

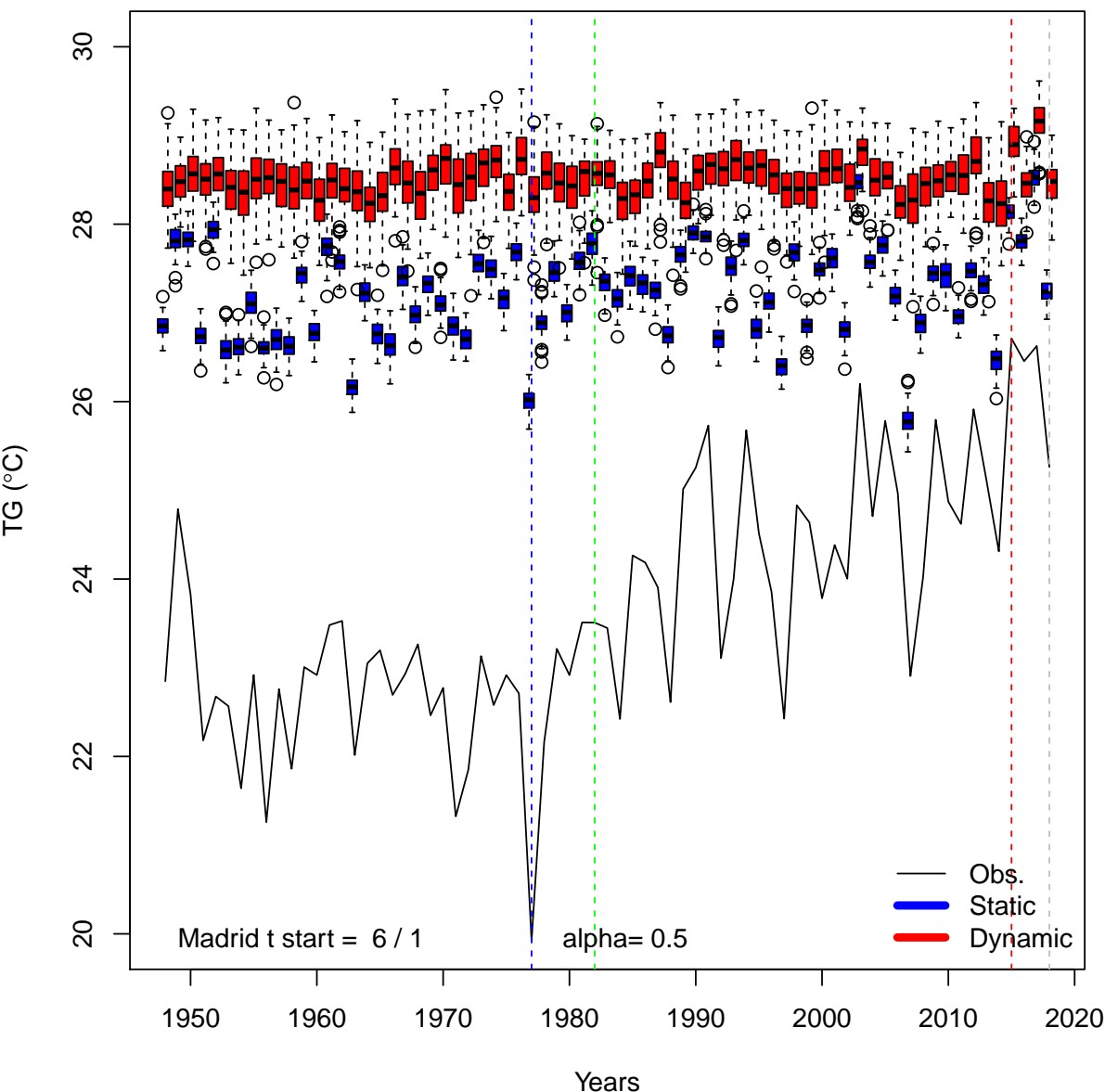

**Figure A5.** Time variations of probability distributions of simulated average temperatures (TG) for $\alpha_{TG} = 0.5$ in Madrid. The black continuous line represents the observed variations of Orly summer averages between 1948 and 2018. Temperatures are expressed in $^\circ$C. The vertical colored lines outline the coldest (blue), median (green), warmest (red) and 2018 summers. The boxplots represent the ensemble variability of the simulations for each year. The red boxplots are for the dynamic simulations. The blue boxplots are for the static simulations. The boxes of boxplots indicate the median ($q50$), lower ($q25$) and upper ($q75$) quartiles. The upper whiskers indicate $\min[\max(T), 1.5 \times (q75 - q25)]$. The lower whisker has a symmetric formulation. The points are the simulated values that are above or below the defined whiskers.

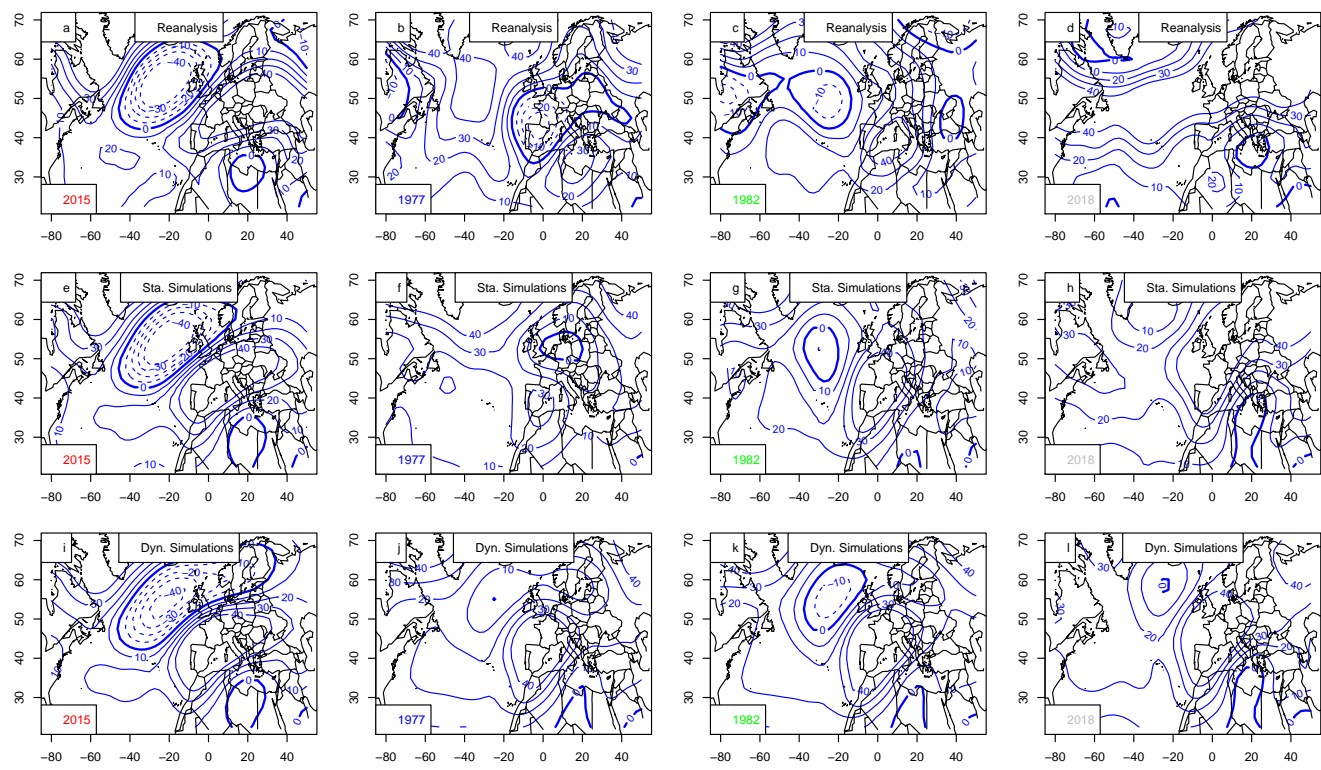

**Figure A6.** Z500 anomaly composites (in m, with 10m increments) for the warmest summer in Madrid (2015), coolest (1977), median (1982) and 2018. The 0 m isoline is indicated with a thick blue line. Negative anomalies are in dashed isolines. Horizontal axes are for longitudes in degrees east. Vertical axes are for latitudes in degrees north. Upper row (a–d): mean Z500 from NCEP reanalyses. Center row (e–h): Static simulations. Bottom row (i–l): Dynamic simulations.

*Author contributions.* PY designed the model and conducted the numerical experiments. AJ stimulated the conception of the manuscript and participated to its writing.

*Competing interests.* The authors declare no competing interest.

*Disclaimer.* TEXT

*Acknowledgements.* We thank F. Bouchet and F. Ragone for enlightening discussions on importance sampling. We thank J. Legrand, N. Legrix, J. Markantonis, P. Pfleiderer, E. Vignotto for discussions on the implementation of the computing code. The two anonymous reviewers greatly helped improving the manuscript. This work was supported by ERC grant No. 338965-A2C2.

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
