# Peer review of "Simulation of Extreme Heatwaves with Empirical Importance Sampling"

_Geoscientific Model Development, 2019_

## Referee Comment (RC1) · Anonymous Referee #1 · 8 Sep 2019

**A) GENERAL COMMENTS**

The manuscript represents a weather-generator-type algorithm for simulating extreme events like heatwaves. Using so-called empirical importance sampling in generating the daily sequences of weather, the algorithm can be tailored to produce extremely warm summers much (presumably, several orders of magnitude) more commonly than they would occur in the real world or without the importance sampling. The algorithm is generic and should therefore also be applicable to other types of "long" extreme events such as extended periods of heavy precipitation.

Importance sampling has previously been applied to simulations by dynamical atmospheric models (e.g., Ragone et al. (2017) cited in the manuscript). Compared with this, the use of a stochastic weather generator requires much less computing time, and

therefore potentially allows a much larger number of simulations.

However, I have a major scientific concern about the fidelity of the method. If the motivation is anything else than to find the extremes of the summer (June-July-August) mean temperature and the associated summer mean atmospheric circulation, the realism of the daily time series also matters. However, as shown by Fig. 5, the warmest simulated summers have no seasonality at all, with equally high temperatures occurring from the beginning of June to the end of August. This appears quite unrealistic, since very high temperatures are much less probable (if possible at all) in the beginning and the end of the summer than in July or early August.

The problem likely results from the fact that the method has no strict constraint on the time of year of the circulation analogies, and the warmest trajectories therefore sample days from the height of the summer even in the beginning and the end of the summer. This could be mitigated, though at the cost of reduced sampling space, by only accepting analogue days from a moving window of (e.g.) +/- 15 calendar days around the target day. Alternatively, it might be possible to work with anomalies (removing the mean seasonal cycle before applying the algorithm) or with normalized anomalies of Z500 and T (removing the mean seasonal cycle and dividing by standard deviation), although this is uncharted terrain that might create its own problems.

B) SPECIFIC COMMENTS

1. P5L1. Does this mean the best 20 analogues regardless of the time of the year (cf. general comments)?

2. P5L18-19 and 26-27. I suppose the weight is directly, not inversely proportional to the correlation, i.e., days with higher correlation are more likely chosen as analogues.

3. P5L28-30. Does this procedure lead to a realistic autocorrelation of the daily mean temperatures?

4. P7L10. I assume that this is an addition to the factors 1-2 on P5L24-27, not a

replacement.

5. P7L16-18. Is physical relevance really ensured? See the general comment about the lack of seasonality.

6. P9L26-27. The only way in which this paper currently answers this question ("how likely is the occurrence of an event") is by fitting a normal distribution to the observed JJA mean temperatures (right scale in Figure 4). Would it be possible to refine these estimates based on the SWG approach (cf. Figure 4 in Ragone et al. (2017), cited in the manuscript)?

7. P12L22-24. This interpretation "small perturbations of the atmospheric Z500 structures can add $\approx$ 4K . . ." is dubious, because there is no one-to-one physical relationship between Z500 and surface temperature. Much more likely, the 4 K addition in temperature comes from the tendency of the algorithm to select the warmest days among days with similar Z500 fields. The slight changes in the Z500 anomalies are a side effect of this, but they are not large enough to "cause" the change in surface temperature. Note that, for the whole atmospheric column to be 4 K warmer, the layer between 500 and 1000 hPa should become 80 meters thicker.

8. P12L27-29. How long does the simulation remember its initial conditions? Would there still be a difference in the Z500 fields if they were only averaged over July-August?

C) TECHNICAL COMMENTS AND CORRECTIONS

1. P1L19: ensembles in plural

2. P2L26: linked to

3. P3L1: recalls

4. Figure 3 could be improved by including the values of the alfa parameter in the figure panels. In addition, it would be useful to describe the interpretation of the box plots in the caption (there are several versions around, although some are more common than

others).

5. Figures 6, 7, A2, A4 and A6. These maps could be improved by using different colors for positive and negative anomalies. It would also be better to use the same contour interval in all panels of each figure.

---

## Referee Comment (RC2) · Anonymous Referee #2 · 20 Sep 2019

**GENERAL COMMENTS**

The authors introduce a new method to simulate extreme heatwaves by using a stochastic weather generator, which is adapted to simulate high temperature values with low probability based on importance sampling. Ragone at al. (2017) had shown that importance sampling can be used to simulate heatwaves with numerical models at low computational costs. The present work is based on the same idea, but the importance sampling is implemented in case of a stochastic weather generator. The authors underline the computational effectiveness and flexibility of their method, which can be implemented to simulate also other types of persistent extreme events.

Although I think that the method is a promising complimentary tool to simulate persistent events, I am afraid that the simulated time series become physically less and less

realistic with increasing alpha values. There are obviously limitations of this method which should be handled more carefully.

Overall, the structure of the manuscript is clear, the abstract is clear and concise. I appreciate that the authors mention several caveats of the method and of the used data sets. Nonetheless, the description of the method is not totally and unequivocally clear. Furthermore, the majority of the figures is hard to read, and some figure captions are lacking important information.

SPECIFIC COMMENTS

P2 L9-12: The statement about EVT is too general and one-sided. EVT has been used successfully to estimate extreme temperature and precipitation events also in case of relatively short time series of about 30 years (see for example Zahid et al. 2017), and has the advantage to provide estimates for unobserved events. It is true, however, that EVT is more useful to model instantaneous extremes instead of long lasting events. Thus, the main problem of using EVT to simulate heat waves lies in the temporal persistence of these events.

P5 L28-29: A more detailed formulation would help understanding.

Sec. 3: In Sec. 3.2 K best analogues are mentioned (with K=20), and in Sec. 3.3 as well. Furthermore $t^{(k)}$ (k as superscipt) is used to denote the dates of the K best analogues. However, in Fig. 2 N analogues are mentioned (N=?) and we find $t^{(1)}$, $t^{(i)}$ and $t^{(90)}$. It seems like the notation is not totally consistent.

P16 L6-8: Other climate variables, like precipitation and wind speed, are very different from temperature in terms of their probability distribution and the auto-correlation. Are the authors totally convinced about the applicability of the presented method also in these cases?

TECHNICAL CORRECTIONS

FIG3: error bars and circle markers are not explained in the caption. In the upper left

panel the black line representing the observations is almost invisible.

FIG4: the meaning of the colours red and blue, the error bars and circle markers are not explained in the caption.

FIG5: What is BDOTM? Orange dashed line is almost invisible.

P2-L14: poor English P2-L26: linked to P3-L1: Sec. 3 recalls P14-L2: The optimal Z500 patterns. . . are similar to...

―――――――――――――――――

---

## Author Comment (AC1) · 15 Nov 2019

**Reply to referee #1**

**A) GENERAL COMMENTS**

The manuscript represents a weather-generator-type algorithm for simulating extreme events like heatwaves. Using so-called empirical importance sampling in generating the daily sequences of weather, the algorithm can be tailored to produce extremely warm summers much (presumably, several orders of magnitude) more commonly than they would occur in the real world or without the importance sampling. The algorithm is generic and should therefore also be applicable to other types of "long" extreme events such as extended periods of heavy precipitation.

Importance sampling has previously been applied to simulations by dynamical atmospheric models (e.g., Ragone et al. (2017) cited in the manuscript). Compared with this, the use of a stochastic weather generator requires much less computing time, and therefore potentially allows a much larger number of simulations. However, I have a major scientific concern about the fidelity of the method. If the motivation is anything else than to find the extremes of the summer (June-July-August) mean temperature and the associated summer mean atmospheric circulation, the realism of the daily time series also matters. However, as shown by Fig. 5, the warmest simulated summers have no seasonality at all, with equally high temperatures occurring from the beginning of June to the end of August. This appears quite unrealistic, since very high temperatures are much less probable (if possible at all) in the beginning and the end of the summer than in July or early August. The problem likely results from the fact that the method has no strict constraint on the time of year of the circulation analogies, and the warmest trajectories therefore sample days from the height of the summer even in the beginning and the end of the summer. This could be mitigated, though at the cost of reduced sampling space, by only accepting analogue days from a moving window of (e.g.) +/- 15 calendar days around the target day. Alternatively, it might be possible to work with anomalies (removing the mean seasonal cycle before applying the algorithm) or with normalized anomalies of Z500 and T (removing the mean seasonal cycle and dividing by standard deviation), although this is uncharted terrain that might create its own problems.

Referee#2 had a similar comment. We thank both of them for their careful reviews. The weight that is devoted to the calendar day was indeed too weak. We were deceived by the experiments of Ragone et al. (2017) of perpetual summer. We now use a "stiffer nudging" for calendar days, and obtain more realistic summers (i.e., summers that eventually end into a Fall season). The procedure is explained below.

**B) SPECIFIC COMMENTS**

1. P5L1. Does this mean the best 20 analogues regardless of the time of the year (cf. general comments)?
Yes, indeed. Then, they are weighed by their calendar day.

2. P5L18-19 and 26-27. I suppose the weight is directly, not inversely

proportional to the correlation, i.e., days with higher correlation are more likely chosen as analogues.

This is true. Please note that this condition is not used in the importance sampling SWG. This is clarified in the revised text.

3. P5L28-30. Does this procedure lead to a realistic autocorrelation of the daily mean temperatures?

This was checked in the original paper of Yiou (GMD, 2014, Figure 4). The analog stochastic weather generators generally underestimate temporal auto-correlation. "Real" temperature is not as random as a stochastic weather generator.

4. P7L10. I assume that this is an addition to the factors 1-2 on P5L24-27, not a replacement.

Thank you for this remark. This rule actually replaces the weight on the spatial correlation, and the weights on the calendar days are maintained. This is clarified in the revised manuscript.

5. P7L16-18. Is physical relevance really ensured? See the general comment about the lack of seasonality.

This is indeed a crucial point (see reply to general comment). This problem was treated by increasing the weight on the calendar days. This "calendar nudging" parameter was estimated by trial and error, by taking the smallest value for which most (e.g. more than 70%) of the dynamic simulations end with dates after the second half of August. In our case (summer temperature simulations), a parameter value of 5 is deemed reasonable for summer temperature simulations. All the figures are changed accordingly. This will be explained in the revised text. Note that if this criterion is used (more than 70% of simulations end towards the end of the season), then the value of this weight parameter might be different for another season. This will be explained in the revised manuscript, with an additional figure (below) showing the dependence to the calendar weight of the percentage of simulations for which the last day is within the last two weeks of the summer.

[Figure]

6. P9L26-27. The only way in which this paper currently answers this question ("how likely is the occurrence of an event") is by fitting a normal distribution to the observed JJA mean temperatures (right scale in Figure 4). Would it be possible to refine these estimates based on the SWG approach (cf. Figure 4 in Ragone et al. (2017), cited in the manuscript)?

This is a good question on the theory. One can give a heuristic formula for the probability of trajectories from the analogues. For example, let A be the smallest number (of analogues) for which

$\frac{1}{S}\sum_{k=1}^{A}\exp(-\alpha_T k) > 1 - \epsilon$, where $\epsilon$ is a small positive number (for example 1/Number of simulations that are needed to observe an event)  and S is the sum of all exponential weights. Then the probability of trajectories is close to $(\frac{A}{K})^M$, where M is the number of independent  days in the season (M=18 for a season of 90 days) and K is the number of analogues (K=20). In this case, we verified that such an estimate is close to the Gaussian quantiles, for values of $\alpha_T$ that are not too large (< 1), if K=20 analogues are considered. Such a formulation needs several parameter adjustments (e.g., $\epsilon$ and M). A rigorous formulation would be a statistics paper in itself. In the case of seasonal temperature averages, it is probably wiser to stick to the Gaussian approximation, which does not require too many trials. This will be mentioned in the revised manuscript.

7. P12L22-24. This interpretation "small perturbations of the atmospheric Z500 structures can add≈4K..." is dubious, because there is no one-to-one physical relationship between Z500 and surface temperature. Much more likely, the 4 K addition in temperature comes from the tendency of the algorithm to select the warmest days among days with similar Z500 fields. The slight changes in the Z500 anomalies are a side effect of this, but they are not large enough to "cause" the change in surface temperature. Note that, for the whole

atmospheric column to be 4 K warmer, the layer between 500 and 1000 hPa should become 80 meters thicker.

Point taken. What we meant was that a small modification of Z500 patterns (a few meters at most) can be associated to large surface temperature changes (4K, i.e. larger changes than the expected isostatic change). This is larger than the temperature trend (~0.1K/decade). This is clarified in the revised manuscript.

8. P12L27-29. How long does the simulation remember its initial conditions? Would there still be a difference in the Z500 fields if they were only averaged over July-August?

The "regular" stochastic weather generator remembers the initial conditions (for temperature) up to one month ahead (Yiou and Déandréis, GMD, 2019). Starting all simulations in June, the SLP/Z500 fields averaged over July-August (rather than JJA) do quantitatively change, although the patterns are still anticyclonic over Europe (see figures below).

**C) TECHNICAL COMMENTS AND CORRECTIONS**

1. P1L19: ensembles in plural

OK.

2. P2L26: linked to

OK.

3. P3L1: recalls

OK.

4. Figure 3 could be improved by including the values of the alpha parameter in the figure panels. In addition, it would be useful to describe the interpretation of the box plots in the caption (there are several versions around, although some are more common than others).

The caption for boxplot figures will be completed in the revision (see comment of referee#2).

5. Figures 6, 7, A2, A4 and A6. These maps could be improved by using different colors for positive and negative anomalies. It would also be better to use the same contour interval in all panels of each figure.

The contour lines use the same interval increments. The lines will be continuous for positive anomalies, dashed for negative anomalies, and thick for 0. This facilitates reading the figure on a B&W printout. An example of maps is given below (Z500 anomalies):

[Figure]

---

## Author Comment (AC2) · 15 Nov 2019

**Reply to Referee #2**

**GENERAL COMMENTS**

The authors introduce a new method to simulate extreme heatwaves by using a stochastic weather generator, which is adapted to simulate high temperature valueswith low probability based on importance sampling. Ragone at al. (2017) had shown that importance sampling can be used to simulate heatwaves with numerical models at low computational costs. The present work is based on the same idea, but the importance sampling is implemented in case of a stochastic weather generator. The authors underline the computational effectiveness and flexibility of their method, which can be implemented to simulate also other types of persistent extreme events. Although I think that the method is a promising complimentary tool to simulate persistent events, I am afraid that the simulated time series become physically less and less realistic with increasing alpha values. There are obviously limitations of this method which should be handled more carefully. Overall, the structure of the manuscript is clear, the abstract is clear and concise. I appreciate that the authors mention several caveats of the method and of the used data sets. Nonetheless, the description of the method is not totally and unequivocally clear. Furthermore, the majority of the figures is hard to read, and some figure captions are lacking important information.

Thank you for the general comments. Most of those comments were also made by the first referee (physical realism of the simulations, figure captions, clarity of the method presentation). A better set of parameters was used to make more realistic simulations (i.e. with a seasonality). This is explained in the text and reflects in revised figures. The figure captions now fully describe the features of the figures. Many points on the methodological presentation were expanded and clarified.

**SPECIFIC COMMENTS**

P2 L9-12: The statement about EVT is too general and one-sided. EVT has been used successfully to estimate extreme temperature and precipitation events also in case of relatively short time series of about 30 years (see for example Zahid et al. 2017), and has the advantage to provide estimates for unobserved events. It is true, however, that EVT is more useful to model instantaneous extremes instead of long lasting events. Thus, the main problem of using EVT to simulate heat waves lies in the temporal persistence of these events.

Fair enough. The statement will be reformulated.

P5 L28-29: A more detailed formulation would help understanding.

OK. We clarified the formulation.

Sec. 3: In Sec. 3.2 K best analogues are mentioned (with K=20), and in Sec. 3.3 as well. Furthermore t(k) (k as superscript) is used to denote the dates of the K best analogues. However, in Fig. 2 N analogues are mentioned (N=?) and we find t(1), t(i) and t(90). It seems like the notation is not totally consistent.

The notation will be streamlined: K instead of N in Fig. 2, and care taken to superscripts. See revised figure below.

[Figure]

Goal: Simulate ensembles of sequences of analogue TG with highest possible temperature and compatible Z500

P16 L6-8: Other climate variables, like precipitation and wind speed, are very different from temperature in terms of their probability distribution and the auto-correlation. Are the authors totally convinced about the applicability of the presented method also in these cases?
Prolonged episodes of precipitation are being tested. We leave this to another paper. Short lived events (e.g. storms or Mediterranean events) are out of the scope of this methodology, as pointed out in the manuscript.

**TECHNICAL CORRECTIONS**

FIG3: error bars and circle markers are not explained in the caption. In the upper left panel the black line representing the observations is almost invisible.
Sorry. Red is for "dynamic" weather generator and blue is for "static" weather generator. The boxplots indicate the median, 25th and 75th quartiles. The upper "whisker" classically indicate min(1.5 (q75-q25)+ q50, max(T)). The lower whisker is the symmetrical formulation. This is explicit in the revised manuscript.

FIG4: the meaning of the colours red and blue, the error bars and circle markers are not explained in the caption.
Sorry. Red is for "dynamic" weather generator and blue is for "static" weather generator. The boxplots indicate the median, 25th and 75th quartiles. The upper "whisker" classically indicate min(1.5 (q75-q25)+ q50, max(T)). The lower whisker is the symmetrical formulation.

FIG5: What is BDOTM? Orange dashed line is almost invisible.
Berlin-De Bilt-Orly-Toulouse-Madrid. This is now explicit in the figure caption. The orange line will be thicker.

P2-L14: poor English
OK. The sentence is rephrased more linearly.

P2-L26: linked to P3-L1: Sec. 3 recalls
OK.

P14-L2: The optimal Z500 patterns...are similar to...
OK.

---

## Author Response (AR2)

Dear Authors,

I have gone through your corrections and modified manuscript. There are a few technical points that need to be considered before accepting the manuscript to GMD. See list here:

Our replies are in blue. A pdf with track change (latexdiff) is attached below.

P2, L10: Simulate should be simulated

No : EVT is useful to investigate AND to simulate events… (here, "simulate" is a verb, not an adjective).

P2, L18: Models should be model

OK.

P2, L25: The reference should be after bias

OK.

P5, L3: degree mark missing from W and E

OK.

P5, L6: Webpages should not be provided as web links. As there is already the reference I suggest to remove the whole link. There is extra bracket after the reference

OK.

P6, L26: comma missing after the equation

OK.

P7, L11: have should be has

OK.

Figure 1: Units (degree) are missing from x- and y-axes

The figure caption now explains that longitudes are expressed in degrees east, and latitude are expressed in degrees north. A random sampling of 5 papers just published in GMD shows maps without such explanations. We are not sure that such a correction is necessary. If those are Copernicus guidelines, they should be clarified in the instructions to authors.

Figure 2: The figure needs to be made more clear. There is a) on the right hand side figure but no label on the left subplot. The different parts of the figure in not appropriate explained in the figure explanation. TG and BDOTM not opened in figure explanation. The units of TG missing from right-side figure. Also x-axis legend is missing.

The figure was modified.

Figure 4: TG not explained and degree missing from the figure. a-d missing from the figure and figure caption.

The definition of TG is given in the first sentence of the temperature observation subsection. The instructions to authors of GMD state that:

**"Figure captions**: Each illustration should have a concise but descriptive caption. The abbreviations used in the figure must be defined, unless they are common abbreviations or have already been defined in the text. Figure captions should be included in the text file and not in the figure files."

Labels (a-d) were added to each panel.

Figure 5: y-axis explanation with units should be added to the figure. Degree missing in from of C on y-axis. TG not explained in the figure explanation.

TG is average daily temperature. This is stated at the beginning of the temperature observation subsection. (see GMD instruction in comment above). °C are added to the y-axes.

JJA not opened.

This is now stated in the first paragraph of the temperature observation subsection.

Figure 6: "panel" can be removed from panel a, panel b,…

OK.

TG not explained and its units are missing.

TG is defined in the text. Units are explained in the caption.

Figure 7-8, A2, A4, A6 : SLP needs to be opened in figure explanation,

SLP is defined at the beginning of the atmospheric circulation subsection and is a standard acronym in atmospheric sciences. See comment above on GMD instructions to authors.

x and y-axis variables and units are missing

Like Figure 1, longitudes and latitudes are in degrees.

Figure A1, A3, A5,: TG not explained and degree missing from the figure. Box plots not explained.

Boxplots are explained in the caption of figure 4. Their definition does not change.

In the code and data availability section, instead of just marking the websites, the correct bibliography entry should be retrieved from the Zenodo entry itself and be included in the bibliography. This should be cited from the section.

OK.

[revised manuscript text omitted]